# GIV/Girdin, a non-receptor modulator for Gαi/s, regulates spatiotemporal signaling during sperm capacitation and is required for male fertility

Sequoyah Reynoso[1†], Vanessa Castillo[2†], Gajanan Dattatray Katkar[2†], Inmaculada Lopez-Sanchez[3], Sahar Taheri[4], Celia Espinoza[2], Cristina Rohena[2], Debashis Sahoo[4,5,6], Pascal Gagneux[1]*, Pradipta Ghosh[2,3,5,7]*

[1]Department of Pathology, School of Medicine, University of California San Diego, San Diego, United States; [2]Department of Cellular and Molecular Medicine, School of Medicine, University of California San Diego, San Diego, United States; [3]Department of Medicine, School of Medicine, University of California San Diego, San Diego, United States; [4]Department of Computer Science and Engineering, Jacob's School of Engineering, University of California San Diego, San Diego, United States; [5]Moore's Comprehensive Cancer Center, University of California San Diego, San Diego, United States; [6]Department of Pediatrics, School of Medicine, University of California San Diego, San Diego, United States; [7]Veterans Affairs Medical Center, Washington DC, United States

**\*For correspondence:**
prghosh@ucsd.edu (PG);
pgagneux@health.ucsd.edu (PG)

[†]These authors contributed equally to this work

**Competing interest:** The authors declare that no competing interests exist.

**Abstract** For a sperm to successfully fertilize an egg, it must first undergo capacitation in the female reproductive tract and later undergo acrosomal reaction (AR) upon encountering an egg surrounded by its vestment. How premature AR is avoided despite rapid surges in signaling cascades during capacitation remains unknown. Using a combination of conditional knockout (cKO) mice and cell-penetrating peptides, we show that GIV (*CCDC88A*), a guanine nucleotide-exchange modulator (GEM) for trimeric GTPases, is highly expressed in spermatocytes and is required for male fertility. GIV is rapidly phosphoregulated on key tyrosine and serine residues in human and murine spermatozoa. These phosphomodifications enable GIV-GEM to orchestrate two distinct compartmentalized signaling programs in the sperm tail and head; in the tail, GIV enhances PI3K→Akt signals, sperm motility and survival, whereas in the head it inhibits cAMP surge and premature AR. Furthermore, GIV transcripts are downregulated in the testis and semen of infertile men. These findings exemplify the spatiotemporally segregated signaling programs that support sperm capacitation and shed light on a hitherto unforeseen cause of infertility in men.

## Introduction

Mammalian sperm acquire their fertilizing potential after insemination, during the passage through the female reproductive tract. Two key consecutive processes are prerequisites for successful fertilization: (i) sperm must first undergo capacitation, a process that is characterized by progressive acquisition of hypermotility, change in membrane, and phosphorylation status, and (ii) they must later undergo acrosome reaction (AR), a process that is characterized by an exocytotic release of acrosomal enzymes to penetrate the zona pellucida of the egg (*Mayorga et al., 2007*; *Hirohashi and Yanagimachi, 2018*). Although capacitation is an important physiological prerequisite before spermatozoa can fertilize the oocyte in every mammalian species studied, the molecular mechanisms and signal

transduction pathways involved in this process are poorly understood. AR, on the other hand, is a time-dependent phenomenon that cannot take place prematurely or too late (*Cummins et al., 1986*). Premature spontaneous AR that occurs in the absence of proper stimuli (AR insufficiency) has been associated with idiopathic male infertility (*Tesarik and Mendoza, 1995*).

Being transcriptionally and translationally silent, mature spermatozoa support capacitation and AR relying exclusively on post-translational events, for example, increase in membrane fluidity, cholesterol efflux, ion fluxes resulting in alteration of sperm membrane potential, and an increased protein phosphorylation; the latter represents a very important aspect of capacitation (*Naz and Rajesh, 2004*) (summarized in *Figure 1—figure supplement 1A,B*). Despite these mechanistic insights into sperm capacitation, key gaps in knowledge persist. For example, although it is known that phosphotyrosine intermediates in the sperm tail culminate in the activation of the PI3K→Akt signaling axis, and that such activation is vital for sperm hypermotility, how tyrosine phosphorylation leads to the activation of PI3K remains unknown (*Tan and Thomas, 2014*; *Breitbart et al., 2005*). Similarly, although it is known that Akt-dependent actin polymerization in the sperm tail requires both protein kinase A (PKA) and protein tyrosine phosphorylation, the linker(s) between signaling and actin dynamics remains unidentified (*Breitbart et al., 2005*; *Etkovitz et al., 2007*; *Roa-Espitia et al., 2016*). Finally, how cAMP surge during capacitation is restricted to the sperm tail, such that its levels remain low in the sperm head, and premature AR is avoided, remains a mystery.

Here we show that GIV (a.k.a., *GIRDers of actIN filament*, Girdin; gene: *CCDC88A*), a multimodular signal transducer that straddles both tyrosine-based and G protein→cAMP signaling cascades (*Midde et al., 2015*; *Kalogriopoulos et al., 2020*), is a key player during sperm capacitation. GIV is an ideal candidate to fill some of the knowledge gaps identified above because many of its functional modules that take part in either tyrosine-based or G protein signaling cascades are reversibly modulated by phosphorylation cascades. First, GIV is a substrate of multiple tyrosine kinases (TKs), both receptor (RTKs) and non-receptor TKs (non-RTKs) alike (*Lin et al., 2011*; *Midde et al., 2018*). Both RTKs and non-RTKs phosphorylate two substrate sites within GIV's C-terminus that, upon phosphorylation, directly bind and activate class 1 PI3Ks (*Lin et al., 2011*; *Midde et al., 2018*). The major consequence of such phosphorylation is that GIV serves as a point of convergence for multi-TK-dependent PI3K signaling. Second, as a bonafide enhancer and a substrate of Akt (*Enomoto et al., 2005*), GIV binds and depolymerizes actin, and in doing so, serves as the only known substrate of Akt that links the PI3K→Akt cascade to cytoskeletal remodeling (*Enomoto et al., 2005*). Third, as a guanine nucleotide-exchange modulator (GEM) for trimeric GTPases, GIV serves as a *g*uanine nucleotide *e*xchange *f*actor (GEF) for Gi (*Garcia-Marcos et al., 2009*) and a *g*uanine nucleotide *d*issociation *i*nhibitor (GDI) for Gs (*Gupta et al., 2016*) via the same evolutionarily conserved C-terminal motif. The major consequence of such versatility of modular function is that by activating the inhibitory Gi and inhibiting the stimulatory Gs proteins GIV overall inhibits membrane adenylyl cyclase (mACs) and suppresses cellular cAMP (*Getz et al., 2019*). 'Free' Gβγ that is released from both classes of Gi/s trimers further enhances the PI3K→Akt signals (*Garcia-Marcos et al., 2009*). We show here how GIV orchestrates distinct spatiotemporally segregated signaling programs in sperm to support capacitation and concomitantly inhibit premature AR, thereby playing an essential role in male fertility.

## Results and discussion
### GIV is highly expressed in spermatocytes

At the time of its discovery in 2005, full-length GIV protein was found to be most highly expressed in two organs: testis and brain (*Figure 1A*). Immunohistochemical studies curated by the Human Protein Atlas further confirm that GIV is most highly expressed in the testis (*Figure 1—figure supplement 2A*). Single-cell sequencing (*Figure 1B*; *Figure 1—figure supplement 2B–E*) and immunohistochemistry (IHC; *Figure 1C*) studies on human testis pinpoint sperm as the major cell type in the testis that expresses GIV mRNA and protein. We confirmed by confocal immunofluorescence on mouse testis that GIV is indeed expressed in the spermatozoa and localizes predominantly to the acrosomal cap, as determined by colocalization with the mouse acrosomal matrix protein, sp56 (*Kim et al., 2001*) (tGIV; *Figure 1D*). As expected, a tyrosine phosphorylated pool of GIV (pYGIV), however, localized mostly to the plasma membrane (PM) (*Figure 1E*). Both antibodies detected the endogenous GIV protein in testicular lysates at the expected size of ~220 kDa (*Figure 1F*). We also noted that GIV

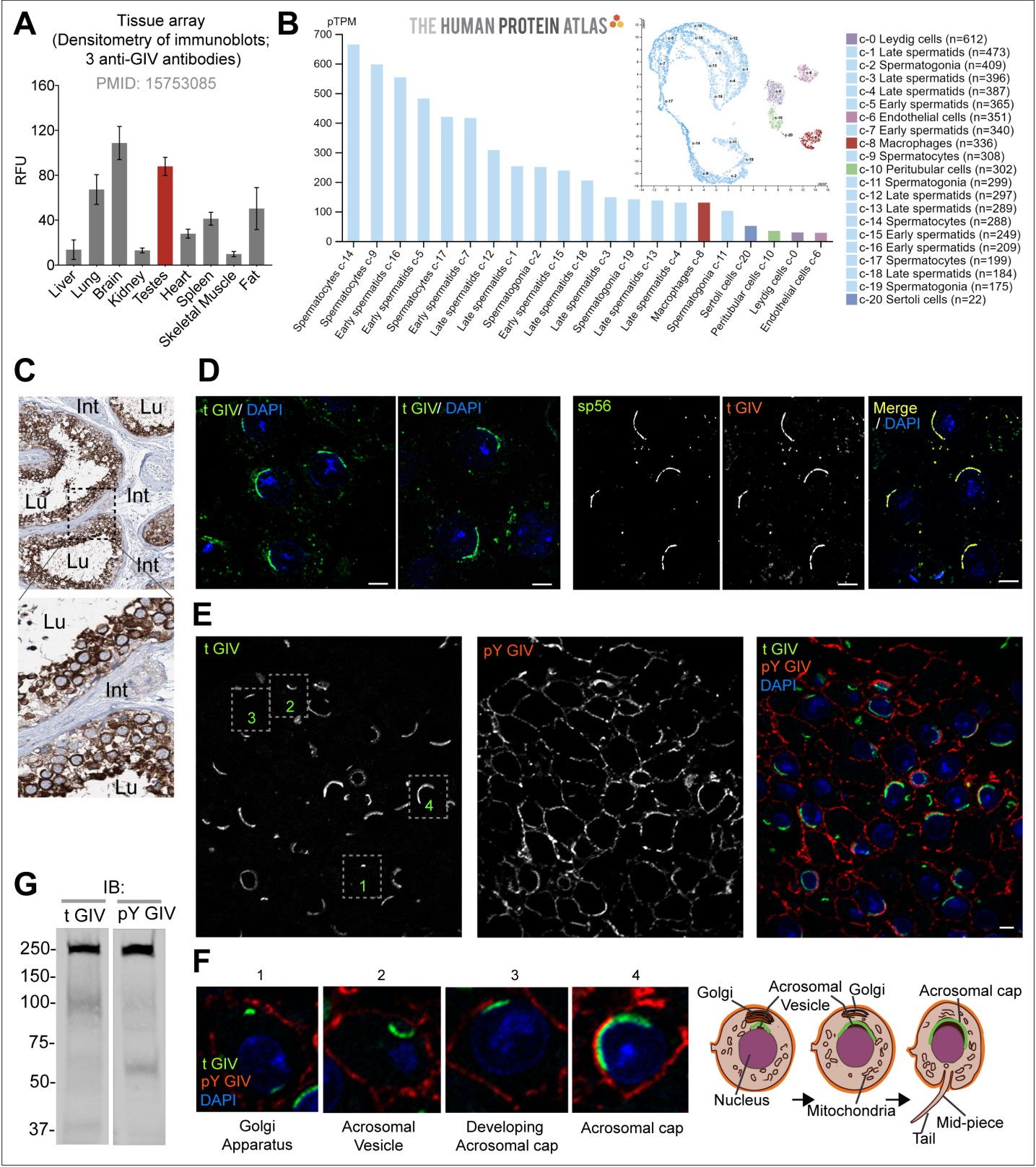

**Figure 1.** GIV (*CCDC88A*) is highly expressed in spermatocytes in testis and localizes to the acrosomal cap. (**A**) Bar graph displays the relative fluorescence unit (RFU) of endogenous full-length GIV protein in immunoblots of organ lysates published previously using three independent anti-GIV antibodies raised against different epitopes of GIV (*Anai et al., 2005*). (*Figure 1—source data 1*)(**B**) RNA expression in the single-cell-type clusters identified in the human testis visualized by a UMAP plot (inset) and a bar plot. The bar plot shows RNA expression (pTPM) in each cell-type cluster.

*Figure 1 continued on next page*

*Figure 1 continued*

UMAP plot visualizes the cells in each cluster, where each dot corresponds to a cell. (**C**) Representative images from human testis immunistochemistry studies curated in the Human Protein Atlas. Int: interstitium; Lu: lumen of seminiferous tubules. (**D**) Cryosections of mouse testis (8 weeks old, C57BL/6) were stained for either total GIV (tGIV; green) and DAPI (blue, nucleus) alone, or co-stained with tGIV and the sperm acrosomal matrix protein zona pellucida 3 receptor (ZP3R, formerly called sp56; red) and analyzed by confocal immunofluorescence. Representative images from two independent analyses are displayed. Scale bar = 10 μm. (**E, F**) Cryosections of mouse testis tissue analyzed for total (t) GIV (green), pY GIV (red), and DAPI (blue, nucleus). Representative images from two independent analyses are shown in panel (**E**); scale bar = 10 μm. Insets in panel (**E**) are magnified and displayed in panel (**F**, left). Schematics in panel (**F**, right) display various localization of GIV observed during the process of maturation of the Golgi into acrosomal cap. (**G**) Immunoblots on mouse testis lysates with the same tGIV and pY GIV antibodies. (*Figure 1—source data 2*).

The online version of this article includes the following figure supplement(s) for figure 1:

**Source data 1.** Quantitative immunoblotting of GIV in tissues.

**Source data 2.** Full-length, uncropped immunoblots on mouse testis lysates with tGIV and pY GIV antibodies (corresponds to *Figure 1G*).

**Figure supplement 1.** Schematic summarizing the known localization (**A**) and role of G proteins/AC proteins and their impact on cAMP signaling (**B**) during sperm processes.

**Figure supplement 2.** *CCDC88A* (GIV/Girdin) is highly expressed in the testes, most specifically the spermatocytes.

consistently and predominantly localizes to the acrosome as it matures from a rudimentary vesicle into a vesicular cap during sperm maturation (*Figure 1G*). GIV's localization to the acrosome, which is derived from the Golgi apparatus (*Khawar et al., 2019*), is in keeping with GIV's predominant localization to the Golgi and Golgi-associated transport vesicles in diverse cell types (*Le-Niculescu et al., 2005*; *Lo et al., 2015*). Taken together, we conclude that GIV is highly expressed in sperm and may be important for sperm functions.

## Transcripts of GIV are reduced in infertile men

Previously, in a publicly available patent (WO2017024311A1), the GIV gene (*CCDC88A*) was identified as one among a panel of genes whose altered expression due to DNA methylation may help diagnose male fertility status and/or the quality of the embryo (*Carrell, 2016*). We asked if the abundance of GIV transcripts in testis or sperm may be altered in infertile men. To this end, we curated all publicly available transcriptomic datasets from the NCBI GEO portal and analyzed them for differences in the abundance of *CCDC88A* transcripts across the annotated (in)fertility phenotypes (*Figure 2A*). *CCDC88A* transcripts were significantly and consistently downregulated in infertile men across all independent datasets analyzed (*Figure 2B–E*), regardless of whether the samples used for transcriptomic studies were testis or sperm. In Klinefelter's syndrome (KS), the most common sex chromosomal disorder in humans that causes primary infertility, reduced *CCDC88A* expression was seen only after puberty and not in pre-pubertal boys (*Figure 2E*). This finding is in keeping with our observation that GIV is most prominently expressed in spermatocytes (*Figure 1*) and that spermatocytes are depleted in KS patients only at the onset of puberty (*Wikström et al., 2004*). Finally, in a study that segregated subfertile from fertile men using commonly used clinical parameters for semen quality, we found that sperm motility, but not concentration or morphology, was the key parameter (*Figure 2F*); when reduced sperm motility was used as a metric of infertility, semen from those subfertile men displayed reduced levels of GIV transcript. These results indicate that reduced GIV expression in testis and sperm is associated with clinically determined male infertility. Given the heterogeneous nature of the datasets (i.e., diagnosed cause of infertility, ranging from genetic syndromes with developmental or hormonal defects to post-chemotherapy to idiopathic), reduced GIV expression could be considered as a shared common molecular phenotype among infertile men.

## GIV is rapidly tyrosine phosphorylated during capacitation

Mature sperm, by virtue of being transcriptionally and translationally inactive, rely entirely upon rapid post-translational modifications to regulate all pre-zygotic processes. Because GIV is a multimodular signal transducer that straddles both tyrosine-based and G protein signaling pathways (*Ghosh, 2016*; *Ghosh, 2015*), we sought to investigate how GIV's functions are altered during sperm capacitation. Because PI3K-Akt signals downstream of TKs is a critical pathway for actin remodeling in the sperm flagellum and for hypermotility (*Tan and Thomas, 2014*; *Breitbart et al., 2005*; *Etkovitz et al., 2007*; *Roa-Espitia et al., 2016*), and GIV serves as a point of convergence for multi-TK-dependent PI3K

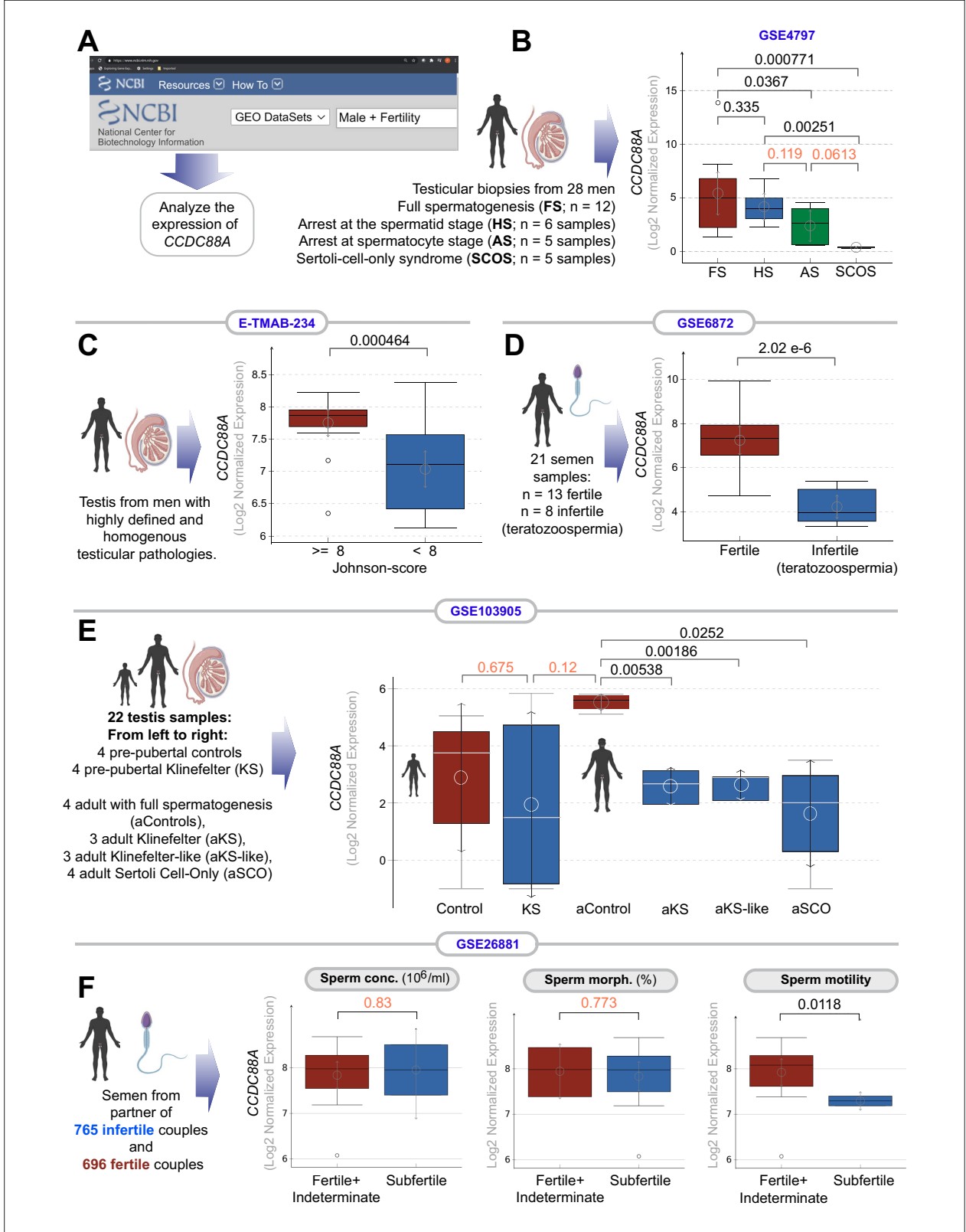

**Figure 2.** Transcripts of *CCDC88A* (GIV) are downregulated in infertile male testis and semen. (**A**) Schematic displays the approach used to search NCBI GEO database for testis and sperm transcriptomic datasets suitable to study correlations between the abundance of *CCDC88A* transcripts and male fertility. (**B–E**) Whisker plots show the relative abundance of *CCDC88A* (expressed as Log2 normalized expression; see Materials and methods for different normalization approaches used for microarray and RNA-seq datasets) in sperm or testis samples (as annotated using schematics) in samples

*Figure 2 continued on next page*

Figure 2 continued

annotated with fertility status, or syndromes associated with infertility. (**F**) Whisker plots show the relative abundance of *CCDC88A* transcripts in sperms classified as subfertile or not based on three properties of sperm assessed using a modified WHO criterion published by *Guzick et al., 2001* (see Materials and methods). Distribution of gene expression values is illustrated using boxplots and mean as circle with 95% confidence intervals (CIs) as arrows. Numbers on top indicate the p values, which were derived from Welch's t-test. A significance level of <0.05, corresponding to 95% CIs are indicated in black font. Insignificant p values are indicated in red font.

signaling (*Lin et al., 2011*; *Midde et al., 2018*), we first asked if GIV is indeed tyrosine phosphorylated in human and mouse sperm during capacitation. Using the sperm swim-up assay, we first confirmed that in human ejaculates low-motile sperm have just as much total GIV as their highly motile counterparts, but by contrast, tyrosine phosphorylated GIV was significantly elevated in the latter (compare tGIV and pYGIV, lanes 1–2 in immunoblots; *Figure 3A*). As a positive control, we simultaneously analyzed the same samples by dual-color immunoblotting with an antibody that detects pan-tyrosine phosphoproteins. As expected (*Ecroyd et al., 2003*; *Ficarro et al., 2003*; *Arcelay et al., 2008*; *Yunes et al., 2003*), the highly motile sperms have higher tyrosine phosphorylation (pan-pY; *Figure 3A*). pYGIV and pan-pY signals co-migrated in the SDS page gel, indicating that GIV is one of the tyrosine phosphorylated proteins in high-motile sperms. The pan-pY and pYGIV signals were found to further increase in capacitated sperms, maximally by 4 hr, without any change in total GIV (lanes 3–4; *Figure 3A*). Such phosphorylation was dependent on the activity of PKA because pretreatment of sperm with the PKA inhibitor H89 virtually abolished both pan pY and pYGIV (*Figure 3B*); these findings are in keeping with the fact that PKA activity is essential for tyrosine phosphorylation cascades during capacitation (*Luconi et al., 2005*; *Lamirande and Gagnon, 2004*). Immunofluorescence studies on human sperm confirmed that pan-pY and pYGIV signals colocalized in the mid-piece and tails of high-motile sperm (*Figure 3C*) where they were significantly induced upon capacitation (*Figure 3D*). Findings in human sperm were mirrored in murine sperm (*Figure 3E,F*), with some notable differences in temporal-spatial dynamics. For example, pY/pYGIV of murine sperms are induced more rapidly and transient. During murine sperm capacitation, pYGIV is induced in 30 min and then reduced in 120 min (*Figure 3F*) and was not as restricted to the sperm tail and mid-piece as in humans (compare sperm head regions in *Figure 3D and F* ). Although full-length GIV (~250 kDa expected size) could be detected in murine sperm (*Figure 3F*), we often detected numerous breakdown products, presumably proteolytic in nature, in both murine and human sperm lysates (*Figure 3A–B and F*). Regardless of the size of the breakdown products, total tGIV, pYGIV, and pan-pY co-migrated in the gels at the same size, suggesting that GIV may be one of the major phosphotyrosine proteins in capacitating sperm. We conclude that GIV is a major phosphotyrosine substrate in sperm tail during capacitation and that its phosphoactivation requires upstream activation of PKA. Our findings suggest that this PKA→T-K→pYGIV axis may enhance PI3K-Akt signals and sperm motility. Because the sperm $Ca^{2+}$ channel, Catsper, exerts both spatial and temporal control over tyrosine phosphorylation as sperm acquire the capacity to fertilize *Chung et al., 2014*, and there is some evidence that H89 may directly inhibit Catsper (*Wang et al., 2020*), the contributions of a possible alternative $Ca^{2+}$→TK→pYGIV pathway towards sperm motility cannot be ruled out.

## The G protein modulatory function of GIV is dynamically phosphoregulated during capacitation

Next, we asked how the G protein modulatory function of GIV is regulated during capacitation. The evolutionarily conserved C-terminal GEM motif in GIV that enables it to both activate Gi (*Garcia-Marcos et al., 2009*) and inhibit Gs (*Gupta et al., 2016*) is phosphoregulated by two Ser/Thr kinases, cyclin-dependent-like kinase 5 (CDK5) (*López-Sánchez et al., 2013*) and protein kinase C Ө (PKCӨ) (*López-Sánchez et al., 2013*) (summarized in *Figure 4A*). Phosphorylation at Ser(S)1,674 induces GIV's ability to activate Gi by ~2.5- to 3.0 -fold, whereas phosphorylation at S1689 inhibits GIV's ability to activate Gi; neither phosphoevent impacts GIV's ability to bind and inhibit Gs. By activating the inhibitory Gi and inhibiting the stimulatory Gs proteins, GIV overall inhibits mACs and suppresses production of cellular cAMP (*Getz et al., 2019*). Because post-translational protein modification is the predominant way mature sperm rapidly respond to environmental cues, we used two previously validated phosphosite-specific antibodies (*López-Sánchez et al., 2013*; *Bhandari et al., 2015*) that detect pS1674-GIV and pS1689-GIV. We found that in mouse (*Figure 4B*, left) and human (*Figure 4C*,

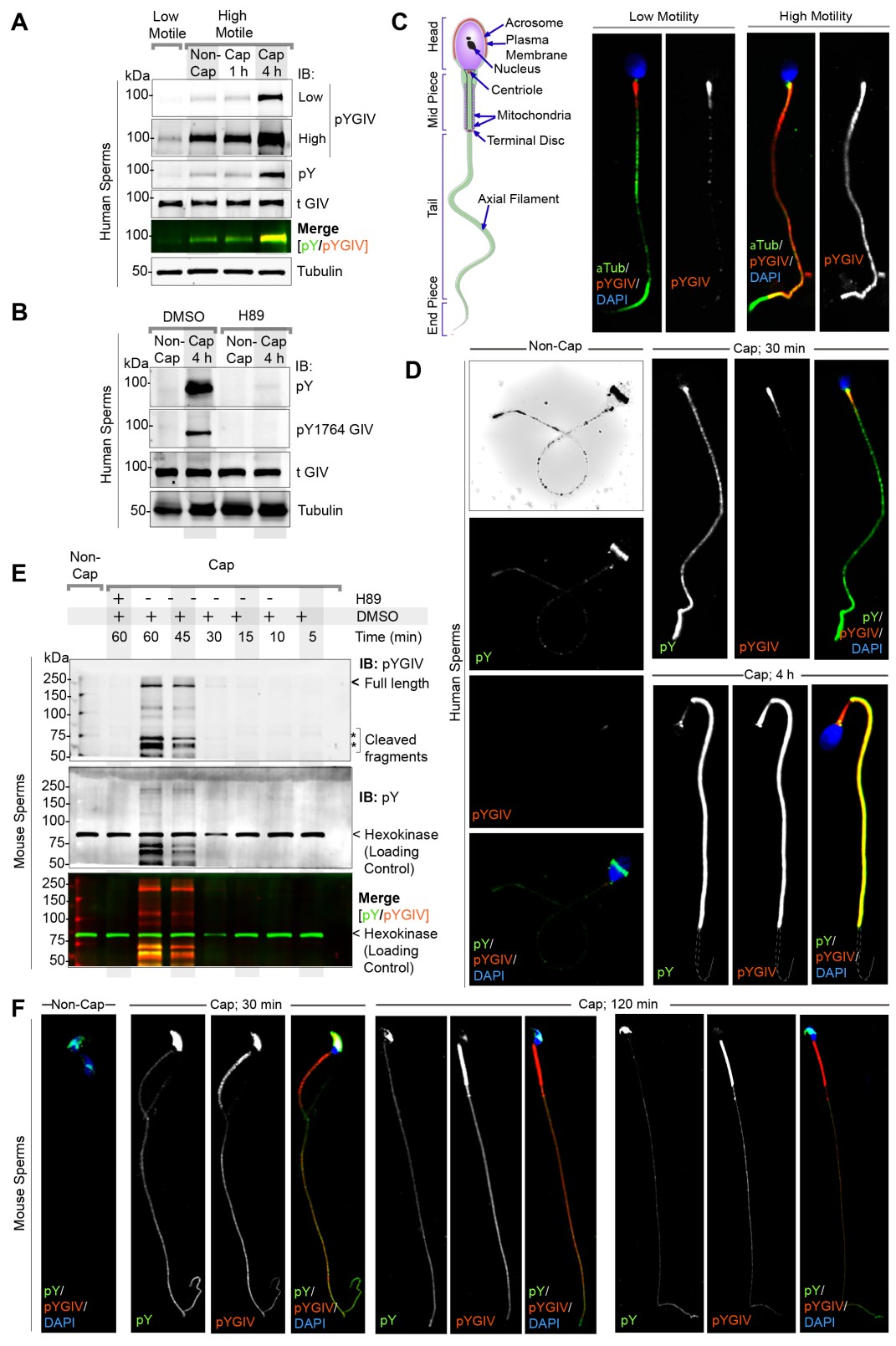

**Figure 3.** GIV localizes to the head and tail of human and murine sperms and is rapidly tyrosine phosphorylated during capacitation. (**A**) Freshly ejaculated human sperm were segregated into low-motile and high-motile populations using 'swim-up' technique (see Materials and methods) and subsequently capacitated in vitro for 1 or 4 hr prior to whole cell lysis. Equal aliquots of lysates were analyzed by immunoblotting for total (t) GIV, pan

*Figure 3 continued on next page*

*Figure 3 continued*

pY, pY1764 GIV (pYGIV), and β-tubulin (loading control) using LI-COR Odyssey (*Figure 3—source data 1*). (**B**) Whole-cell lysates of human sperms capacitated with or without preincubation with H89 (protein kinase A [PKA] inhibitor) or DMSO control were analyzed as in (**A**) (*Figure 3—source data 2*). (**C, D**) Human sperm with low vs. high motility (**C**), were capacitated or not (**D**), fixed and co-stained for total and pY GIV (tGIV; pY GIV), tubulin and DAPI. Representative images that capture the most frequently observed staining patterns (at >80% frequency) among ~100–150 sperms/sample, in three independent samples, derived from three unique subjects are shown. Scale bar = 10 μm. (**E**) Immunoblots of equal aliquots of whole-cell lysates of mouse sperm capacitated with (+) or without (-) pretreatment with PKA inhibitor (H89) or vehicle (DMSO) control. Hexokinase is used as a loading control (*Figure 3—source data 3*). (**F**) Non-capacitated (non-cap) or capacitated mouse sperm were fixed and stained as in (**D**) and analyzed by confocal microscopy. Representative images that capture the most frequently observed staining patterns (at >80% frequency) among ~50–100 sperms/sample, in three independent samples, derived from three mice are shown. Scale bar = 10 μm.

The online version of this article includes the following figure supplement(s) for figure 3:

**Source data 1.** Full-length, uncropped immunoblots on human sperm lysates with tGIV and pY GIV antibodies (corresponds to *Figure 3A*).

**Source data 2.** Uncropped immunoblots on human sperm lysates with tGIV and pY GIV antibodies (corresponds to *Figure 3B*).

**Source data 3.** Full-length, uncropped immunoblots on sperm lysates with pan-pY and pY GIV antibodies (corresponds to *Figure 3E*).

---

left) sperm, HCO3⁻-induced capacitation induced the levels of phosphorylation at the activation site pS1674 in the sperm tails of both species, with two notable inter-species differences: (i) in murine sperm, the acrosomal cap showed phosphorylation at baseline with no further increase upon capacitation; and (ii) in human sperm, the mid-piece region showed phosphorylation at baseline with no further increase upon capacitation. Unlike the activating pS1674 site, distribution/intensity of phosphorylation at the inhibitory pS1689 site was observed at baseline in the head, mid-piece, and tail of the murine sperm (*Figure 4B*, right), and the mid-piece and tail in human sperm (*Figure 4C*, right) and did not change during capacitation.

We next repeated the studies with the sequential addition of HCO3⁻ (for 2 hr) followed by two other stimuli that are commonly used to trigger the AR, the calcium ionophore, A23186 (*Tateno et al., 2013*), and the reproductive hormone, progesterone (*López-Torres and Chirinos, 2017*). To monitor the phosphomodifications in GIV and their temporal relationship with acrosome exocytosis, we co-stained the sperm with peanut agglutinin (PNA) and Ser/Tyr-GIV. PNA binds specifically to galactose residues on the outer acrosomal membrane, and its disappearance is a widely accepted method of monitoring acrosome exocytosis (*Mortimer et al., 1987*; *Kallajoki et al., 1986*). pYGIV was induced predominantly in the mid-piece and tail during capacitation (cap 2 hr; *Figure 4D*) as seen before (*Figure 3F*) but also in the sperm head in the presence of A23187 and progesterone (*Figure 4D*). The localization of pYGIV in sperm head was seen only when the acrosomes were intact and lost in those where the acrosome was shed (compare AC-intact vs. -shed; *Figure 4D*). Similarly, phosphorylation at the activation site pS1674 was detected in sperm heads in the presence of A23187 and progesterone, but exclusively when the acrosomes remained intact (compare AC-intact vs. -shed; *Figure 4E*).

Taken together, the predominant findings can be summarized as follows (see legend of *Figure 4F*): GIV-GEM is inactive at baseline and activated upon capacitation. It remains active in both head and tail regions of capacitated sperm until the moment the acrosome is shed. Capacitation is also associated with robust tyrosine phosphorylation of GIV in the sperm tail and mid-piece throughout the process of acrosomal reaction (AR).

## GIV is required for male fertility

To determine if GIV is required for male fertility, we next co-housed female mice with conditional GIV knockout male mice (henceforth referred to as GIV-cKO; generated using tamoxifen in *Ccdc88a*$^{fl/fl}$-*Ubc*$^{Cre-Ert2}$ mice) or control littermates (WT; *Ccdc88a*$^{fl/fl}$ mice) (see Materials and methods; see legend of *Figure 5A*) and analyzed diverse readouts. GIV knockdown was confirmed by genotyping tail tips (*Figure 5B*) and assessing GIV mRNA (*Figure 5C*) and protein (*Figure 5D*) in the testis. We noted

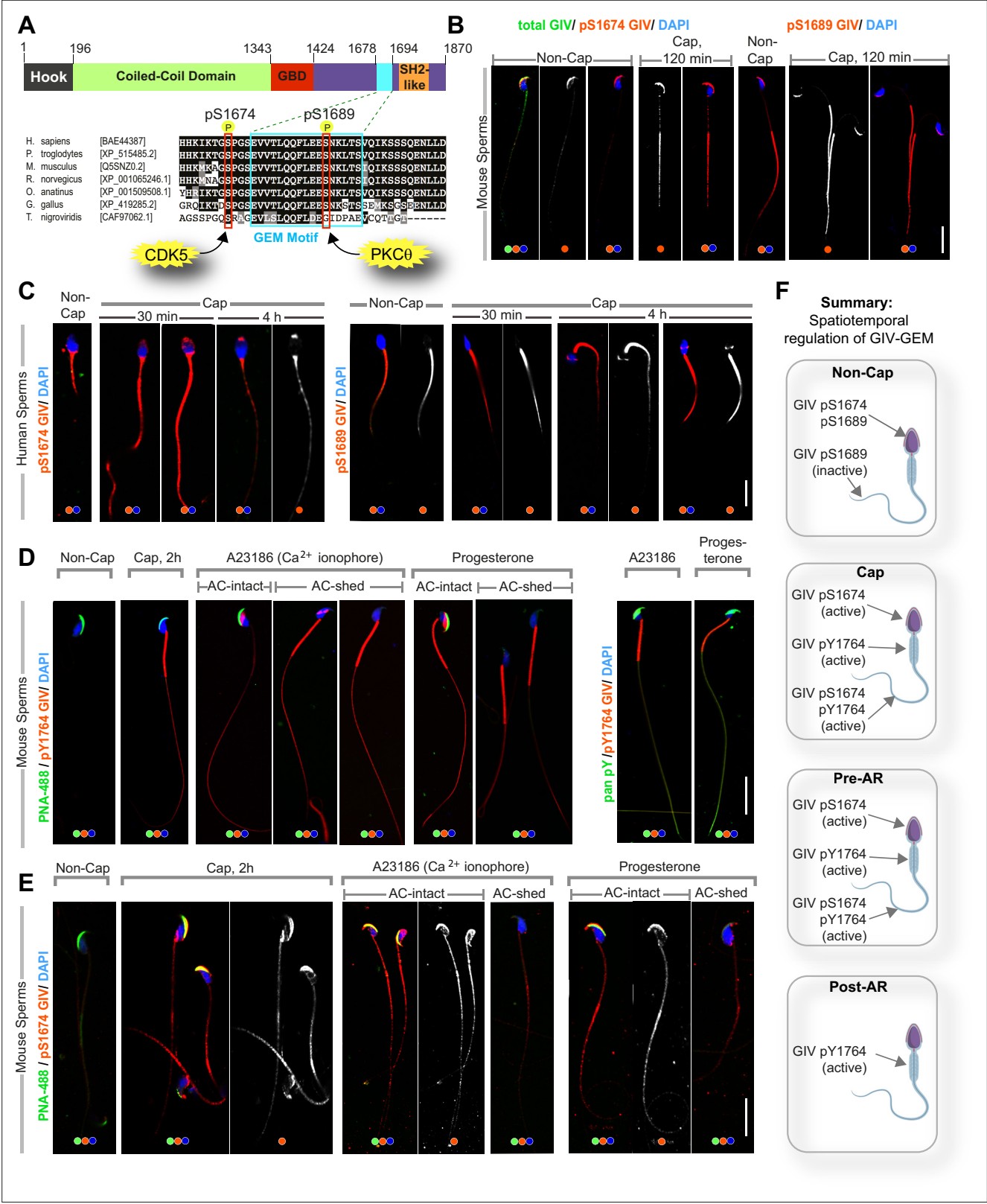

**Figure 4.** GIV's guanine nucleotide-exchange modulator (GEM) function is dynamically phosphoregulated during capacitation and acrosomal reaction (AR) in a spatiotemporally segregated manner. (**A**) Schematic shows the domain map of GIV (top) and the evolutionarily conserved GEM motif within its C terminus. A functional GEM motif is required for GIV to bind and activate Gαi as well as bind and inhibit Gαs (*Gupta et al., 2016*). Important phosphoserine modifications that regulate GIV's GEM motif and the corresponding target kinases are highlighted. (**B, C**) Non-capacitated and

*Figure 4 continued on next page*

Figure 4 continued

capacitated mouse (**B**) and human (**C**) sperm were fixed and analyzed for the phosphoserine modifications highlighted in (**A**). (**D, E**) Mouse sperm with/without capacitation followed by treatment with either Ca²⁺ ionophore or progesterone to trigger AR were fixed and co-stained for peanut agglutinin (PNA-488; green, an acrosomal marker) and either pYGIV (**D**) or pSerGIV (**E**) and DAPI. Representative images are shown. Scale bar = 10 μm. (**F**) Schematic summarizes the spatially segregated phosphomodifications on GIV before and after capacitation and AR in various parts of the sperm. (i) Inhibitory phosphorylation at pS1689 on GIV is seen in both head and tail prior to capacitation (**F**, top); (ii) activating phosphorylation at S1674 on GIV is seen in the sperm head and tail, whereas pYGIV is predominantly seen in the mid-piece and the tail regions upon capacitation (post-cap; **F**) as well as during AR before the acrosome is shed (pre-AR; **F**); and (iii) after the acrosome is shed, pYGIV is the only phospho-GIV that is detected, and predominantly in the mid-piece (post-AR; **F**). Representative images that capture the most frequently observed staining pattern(s) (at >80% frequency), among ~50–150 sperms/sample, three independent samples, derived either from human subjects (n = 3) or mice (n = 3) are shown.

a significant reduction of cumulative probability of pregnancy (100% vs. 55% rate for WT and KO groups, respectively, within 40 days after co-housing; *Figure 5E*) and average litter size (*Figure 5F*) in GIV-cKO mice. Surprisingly, both WT and GIV-cKO mice had similar sperm counts (*Figure 5G*), testes sizes, and weights (*Figure 5—figure supplement 1*). We confirmed by IHC that GIV was predominantly expressed in sperm in the testis of WT mice and that it was effectively depleted in GIV-cKO mice (*Figure 5H*). RNA-seq of the testis followed by unsupervised clustering showed that GIV-cKO testis differentially expressed only a handful of transcripts compared to WT testis (*Figure 5I*). The predominantly upregulated genes mapped to the 'aberrant activation of PI3K/Akt signaling' pathway (*Figure 5J*). This was largely attributable to *Esr1* (highlighted in red; *Figure 5I*); polymorphisms of this gene are known to predispose to male fertility (*Ge et al., 2014*; *Galan et al., 2005*), and its induction represents a negative feedback event, resulting in the setting of inhibition of PI3K signaling (*Bosch et al., 2015*). The predominantly downregulated genes mapped to the IL12 pathway (*Figure 5K*), which is consistent with prior studies in men showing that IL12 may be important for male fertility and that its dysregulation may reflect infertility (*Naz and Evans, 1998*; *Naz et al., 1998*). Notably, both pathways reflect changes that are largely contributed by non-sperm cells in the testis; Esr1 is expressed exclusively in the Leydig cells in mouse testis (*Zhou et al., 2002*; *Kotula-Balak et al., 2005*) and IL12 is largely expressed by endothelial cells, peritubular cells, and macrophages (*Terayama et al., 2014*).

These findings demonstrate that GIV is required for male fertility and suggest that the role of GIV and its various phosphomodifications we observe in sperm is largely post-transcriptional and post-translational in nature.

## GIV's GEM function facilitates hypermotility and survival during sperm capacitation

Next, we assessed the role of GIV during sperm capacitation using a previously validated approach, that is, exogenous addition of cell-permeable His-tagged GIV-derived ~210 aa long peptides (*Ma et al., 2015*); these peptides either have an intact functional GEM motif (WT peptides) or, as negative control, a well-characterized F1685A (FA) mutant of the same motif ,which lacks such activity (*Garcia-Marcos et al., 2009*; *Kalogriopoulos et al., 2019*; *Figure 6A*; top). By anti-His staining followed by flow cytometry, we confirmed that TAT-His-GIV peptides were indeed taken up as we could detect uptake only when staining was conducted under permeabilized conditions (*Figure 6A*, bottom). Peptide uptake was efficient, varying within the range of ~80–90% (*Figure 6A*, bottom). Immuno-fluorescence studies confirmed that uptake was seen in all segments of the sperm (*Figure 6B*). The peptides were detected and functional (i.e., retained their ability to bind Gαi) at 1 and 6 hr post-uptake, as determined using lysates of peptide-transduced sperm as source of GIV in pulldown assays with recombinant GDP-loaded GST-tagged G protein, Gαi3 (*Figure 6C*).

Next, we analyzed phosphoproteins in TAT-GIV-transduced sperm undergoing in vitro capacitation by immunoblotting. Although PKA activation (*Figure 6D*) and pan-Y or pYGIV phosphorylation (*Figure 6E*) were relatively similar between WT and FA-transduced sperm, phosphorylation of Akt differed; TAT-GIV-WT induced phosphorylation of Akt much more robustly than TAT-GIV-FA (*Figure 6D*). This finding is consistent with the established role of GIV-GEM in the activation of the PI3K→Akt pathway via the activation of Gi and the release of 'free' Gβγ15. Because Akt phosphorylation has been implicated in sperm hypermotility and survival during capacitation (*Quan and Liu, 2016*; *Pujianto et al., 2010*), we performed computer-assisted sperm analysis (CASA) and MTT assays, respectively (*Figure 6F*). Consistent with the patterns of Akt phosphorylation, WT, but not FA

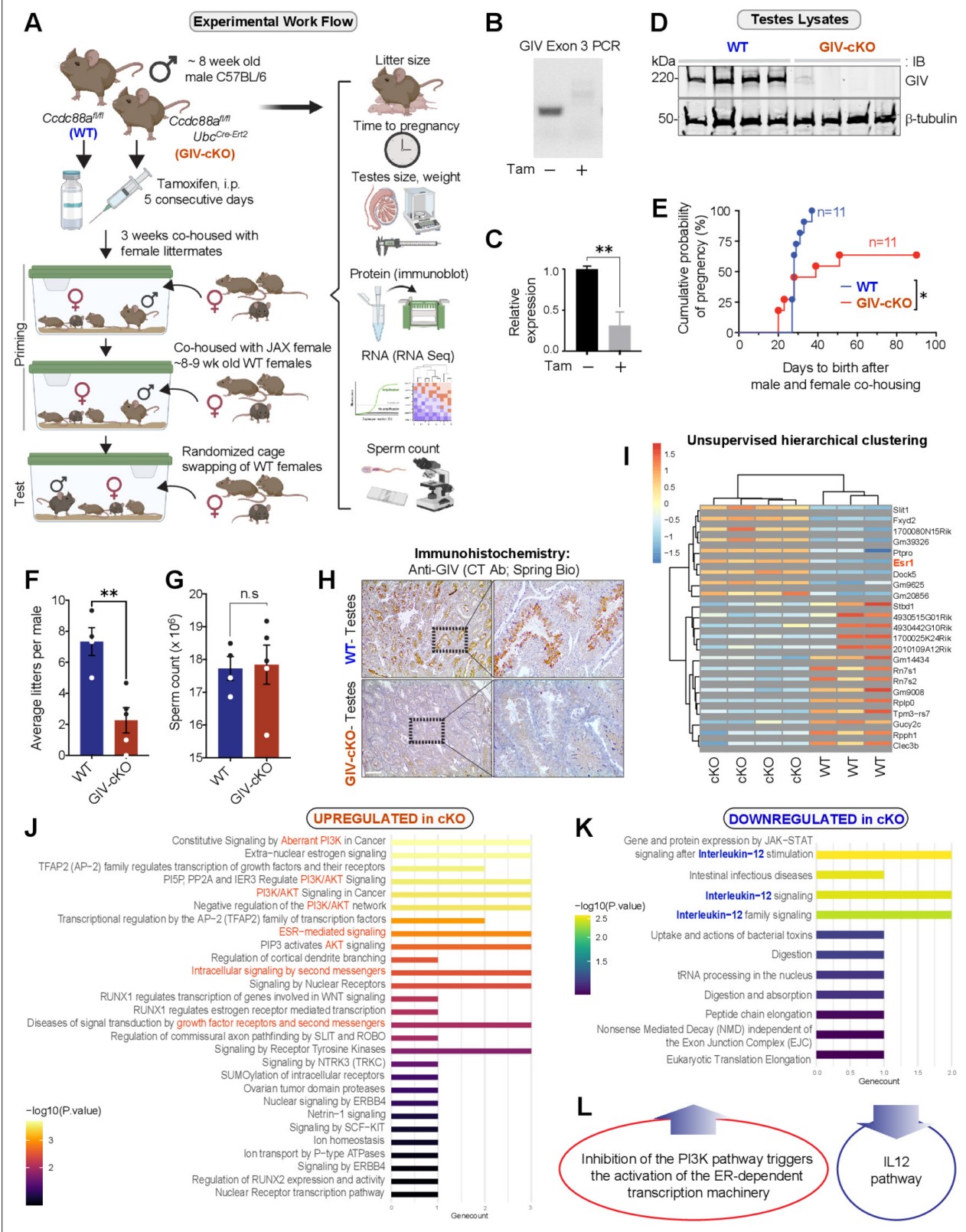

**Figure 5.** GIV is required for fertility in male mice. (**A**) Schematic showing the workflow for fertility studies in conditional GIV-cKO mice. After intraperitoneal injection of tamoxifen, male mice were first primed in two phases—first by co-housing with female littermates × 3 weeks, and subsequently by co-housing with female mice from Jackson laboratory (JAX) while the females acclimatized to the animal facility. The final 'test' group consisted of tamoxifen-injected WT and GIV-cKO male mice randomly assigned to and co-housed with three female mice from JAX, each with

*Figure 5 continued*

proven ability to get pregnant. (**B–D**) Confirmation of GIV-cKO in the mice after tamoxifen injection by genotyping (**B**), qPCR of testis tissues (**C**), and immunoblotting of testis lysates (**D**) (*Figure 5—source data 1*). (**E**) Kaplan–Meier plot showing the cumulative probability of pregnancy (expressed as %) in the females co-housed with either WT or GIV-cKO males. Statistical significance was assessed using log-rank analysis. *p<0.05 (*Figure 5—source data 2*). (**F, G**) Bar graphs showing the average litter size (**F**; *Figure 5—source data 3*) and sperm count (**G**; *Figure 5—source data 4*) in WT and GIV-cKO males. See also *Figure 5—figure supplement 1* for quantifications of tested weight and length. (**H**) Immunohistochemistry staining on mouse testis. Scale bar = 200 µm. (**I**) Unsupervised clustering of WT and KO testis samples based on gene expression. Differentially expressed genes (DEGs) that were up- or downregulated in KO are annotated on the right side (*Figure 5—source data 5*). (**J, K**) Reactome pathway analyses showing the pathways that are up or downregulated in KO testis. (**L**) Summary of the most prominent conclusions from RNA-seq dataset.

The online version of this article includes the following figure supplement(s) for figure 5:

**Source data 1.** Full-length, uncropped immunoblots on testes lysates with GIV and tubulin antibodies (corresponds to *Figure 5D*).

**Source data 2.** Excel sheet with time to live birth values observed in females co-housed with WT and GIV-cKO mice (corresponds to graphs in *Figure 5E*).

**Source data 3.** Excel sheet with litter size values from WT and GIV-cKO mice (corresponds to graphs in *Figure 5F*).

**Source data 4.** Excel sheet with sperm count values from WT and GIV-cKO mice (corresponds to graphs in *Figure 5G*).

**Source data 5.** Excel sheet with differential expression analysis-derived reactome pathway analyses of the most significantly up- and downregulated genes in GIV-cKO mice (corresponds to graphs in *Figure 5I*).

**Figure supplement 1.** GIV is required for male fertility, but its depletion does not impact testes weight or length.

**Figure supplement 1—source data 1.** Excel sheet with testes weight values from WT and GIV-cKO mice (corresponds to the left-side graph in *Figure 5—figure supplement 1*).

**Figure supplement 1—source data 2.** Excel sheet with testes length values from WT and GIV-cKO mice (corresponds to the left-side graph in *Figure 5—figure supplement 1*).

peptide-transduced sperm showed greater overall motility as well as hypermotility (*Figure 6G*) and greater viability (*Figure 6H,I*).

These findings indicate that GIV's GEM function may be dispensable for the PKA→TK→tyrosine phosphorylation pathway, but is required for Akt activation, sperm motility, and survival during capacitation (*Figure 6J*).

## GIV's GEM function suppresses cAMP and AR

Prior studies have underscored the importance of mACs and their role in the regulation of cAMP and acrosome exocytosis in sperm (summarized in *Figure 1—figure supplement 1A*). mACs are localized most abundantly in the head (*Figure 1—figure supplement 1A*), and their activation by Gs or inhibition by Gi is known to finetune cAMP surge in that location, a function that is conserved in numerous species (*Spehr et al., 2004*). Thus, mACs and sACs regulate cAMP surges in the sperm head and tail, respectively, in a spatiotemporally segregated and independent manner (*Figure 1—figure supplement 1B*). Upon approaching the zona pellucida of an egg, a timely surge in cAMP in sperm head is required for the downstream activation of effectors PKA (*Romarowski et al., 2015*) and the exchange proteins directly activated by cAMP (EPAC) (*Sosa et al., 2016*; *Mata-Martínez et al., 2021*), which in turn coordinate the activation of several small GTPases (*Branham et al., 2009*; *Pelletán et al., 2015*; *Bustos et al., 2015*; *Bustos et al., 2012*) of the Ras superfamily. These GTPases enable rapid cytoskeletal remodeling and membrane trafficking events that culminate in acrosome exocytosis. As an activator of Gi and an inhibitor of Gs (*Gupta et al., 2016*) using the same conserved GEM motif (*Figure 7A*), GIV is known to tonically and robustly suppresses cAMP (*Getz et al., 2019*; *Getz et al., 2020*), and by that token, it is expected to inhibit the cAMP surge. Because GIV-GEM was activated upon capacitation and remained active until the acrosome was shed (*Figure 4F*), we hypothesized that GIV's GEM function may be required for the prevention of a premature cAMP surge in the sperm head, and hence, premature acrosome exocytosis. We first confirmed that cAMP is modulated by a variety of stimuli targeting Gi- (adenosine) and Gs-coupled (progesterone) GPCRs (*Figure 7—figure supplement 1A*), consistent with what has been observed before (*Wang et al., 2020*; *Parinaud and Milhet, 1996*). When the same studies were carried out on TAT-GIV peptide-transduced sperm, the expected degree of cAMP induction were observed once again (*Figure 7B*), but TAT-GIV-WT, but not the GEM-deficient FA mutant peptides could significantly suppress the degree of cAMP surge across all stimuli tested (*Figure 7C*, *Figure 7—figure supplement 1B*).

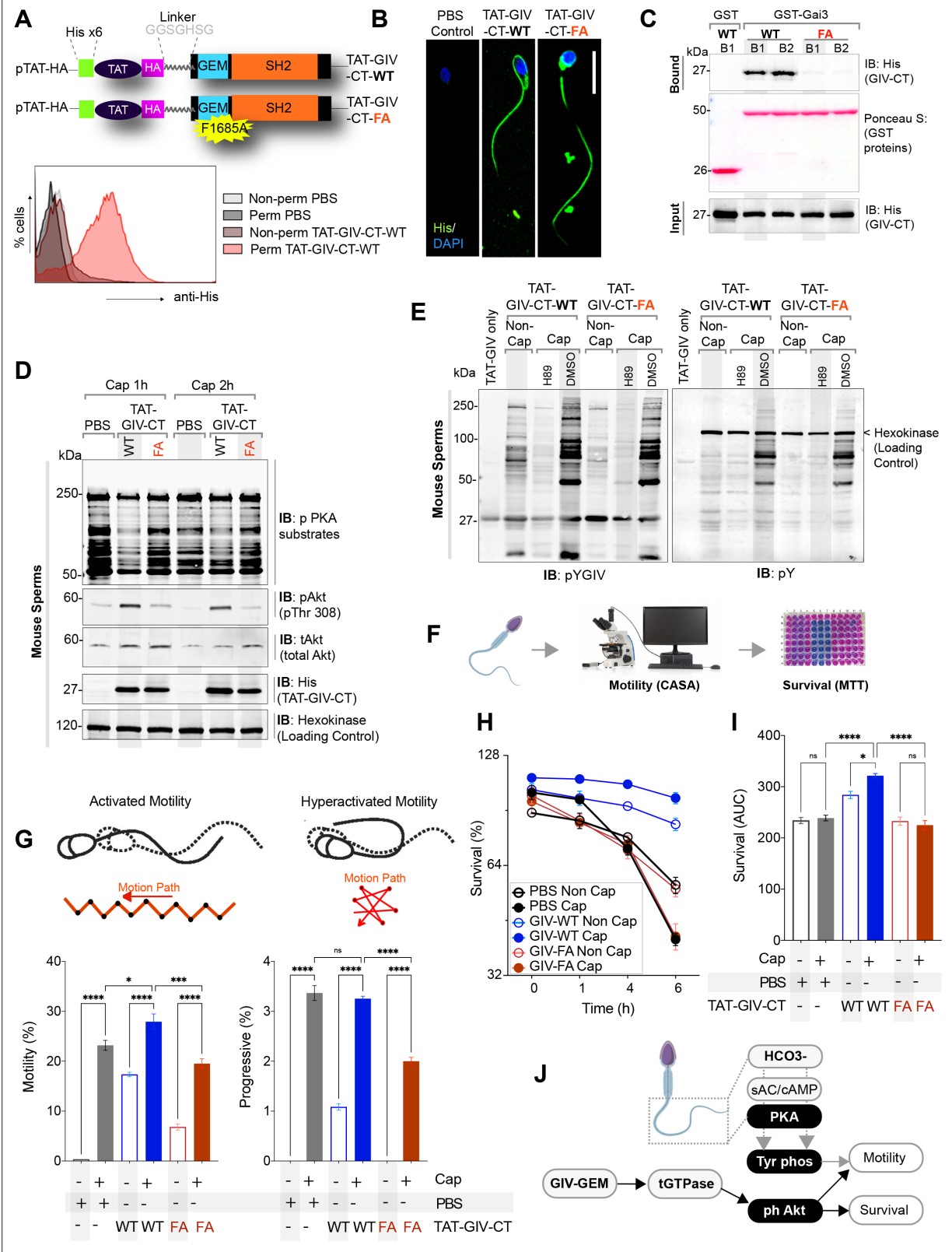

**Figure 6.** GIV's GEM function is required for sperm motility and survival during capacitation. (**A**, **B**) Schematic (**A**, top) of cell-permeant His-TAT-GIV-CT wildtype (WT) and GEM-deficient mutant (F1685A; FA) peptides used in this work. Immunofluorescence images (**B**) representative of sperms after treatment with cell-permeant TAT-GIV-CT peptides and stained with anti-His antibody and DAPI. Scale bar = 15 μm. Histograms (**A**, bottom) from flow cytometry studies conducted with or without permeabilization confirm the uptake of His-TAT peptides in sperm. (**C**) Immunoblots of GST pulldown

*Figure 6 continued on next page*

*Figure 6 continued*

assays testing the ability of GDP-loaded GST-Gαi3 to bind TAT-GIVCT peptides from lysates of sperms at 1 hr (B1) and 6 hr (B2) after transduction (*Figure 6—source data 1*). (**D, E**) Immunoblots of lysates of TAT-GIVCT-transduced sperms at the indicated time points after capacitation analyzed for phospho-PKA substrates (**D**), phospho(p) and total (t) Akt (**D**; *Figure 6—source data 2*), pYGIV (**E**, left), pan-pY (**E**, right) (*Figure 6—source data 3*), and hexokinase (loading control, **D**). (**F–I**) Schematic in (**F**) summarizes workflow in assessing motility and survival of sperms during capacitation. Bar graphs in (**G**; *Figure 6—source data 4*) display the relative % of motile and progressively motile population of sperms. Line graphs in (**H**) show survival of sperms as determined by methy thiazolyl tetrazolium (MTT) assay; bar graphs in (**I**) show the area under the curve (AUC) of the line graphs in (**H**) (*Figure 6—source data 5*). All results are presented as average ± SEM of three independent studies conducted on sperm isolated from three mice. Statistical significance was assessed using one-way analysis of variance (ANOVA) followed by Tukey's test for multiple comparisons. *$p<0.05$, ***$p<0.001$, ****$p<0.0001$, $^{ns}$ $p>0.05$. (**J**) Schematic summarizes the conclusions of how GIV's GEM function impacts sperm phenotypes during capacitation.

The online version of this article includes the following figure supplement(s) for figure 6:

**Source data 1.** Full-length, uncropped immunoblots on GST pulldown assays (corresponds to *Figure 6C*).

**Source data 2.** Full-length, uncropped immunoblots on TAT-peptide-transduced sperm lysates with His, hexokinase, phospho-PKA substrate, phospho-Akt, and total Akt antibodies (corresponds to *Figure 6D*).

**Source data 3.** Full-length, uncropped immunoblots on TAT-peptide-transduced sperm lysates with pan-pY and pYGIV antibodies (corresponds to *Figure 6E*).

**Source data 4.** Excel sheet with sperm motility values (corresponds to graph in *Figure 6G*).

**Source data 5.** Excel sheet with sperm viability values (corresponds to graph in *Figure 6H,I*).

Next we assessed the effect of GIV-GEM on acrosome exocytosis under the same conditions, that is, capacitation followed by AR, as we did in (*Figure 4D,E*), using a highly sensitive immunofluorescence-based assay that monitors the progressive exposure during AR of the inner acrosomal membrane protein, CD46 (*Carver-Ward et al., 1997*; *Frolikova et al., 2016*) (a.k.a. membrane cofactor protein [MCP]; *Figure 7D*). At 15 min after exposure to 10 μM A23186 or 100 μM progesterone, the TAT-GIV-WT, but not TAT-GIV-FA-transduced sperm had more intact acrosomes (*Figure 7E*) and fewer completely reacted acrosomes (*Figure 7F*). These results indicate that AR in response to both A23186 and progesterone was suppressed by TAT-GIV-WT, but not the GEM-deficient FA mutant. Instead, the FA mutant peptide had a higher proportion of sperm that completed AR.

These findings demonstrate that GIV is sufficient to inhibit cAMP surge and AR, and that these functions require a functional GEM module. Taken together with the temporal nature of the GIV-GEM activity (see *Figure 4F*), our findings also suggest that GIV-GEM may inhibit premature cAMP surge and acrosome shedding. Because these premature events may compromise fertilization only in the in vivo setting where sperm is required to remain in capacitated state while maintaining intact acrosomes for prolonged periods of time within the female reproductive tract before encountering the egg, we hypothesized that GIV's function may be bypassed in the setting of in vitro fertilization (IVF; *Figure 7G*). We found this indeed to be the case because TAT-GIV-WT peptide-transduced sperm successfully fertilized the eggs in vitro to a similar extent as PBS control (*Figure 7H,I*). The GEM-deficient FA mutant-transduced sperm, which had higher surges in cAMP (*Figure 7C*) and a higher proportion of completely reacted acrosomes (*Figure 7F*), showed an approximately twofold increase in fertility.

Taken together, these findings indicate that GIV-GEM inhibits cAMP surge and AR to primarily prevent both events from occurring prematurely in vivo until in the presence of an egg for successful fertilization.

## Conclusions

The major discovery we report here is a role of GPCR-independent (hence, non-canonical) G protein signaling in the sperm that is mediated by GIV/Girdin. Expressed most abundantly in the testis, and primarily in sperm, GIV is required for male fertility, and low GIV transcripts in men were invariably associated with infertility. We show that GIV is rapidly phosphomodulated on key tyrosine and serine residues in a manner segregated in space and time in various segments of the sperm (head, mid-piece, and tail) during capacitation and acrosomal reaction. These specific phosphomodifications, which are known to regulate GIV's interactions with other key proteins (PI3K, Gαi/s proteins, etc.) and its functions as an effector of multiple TKs, as a cytoskeletal remodeler, and as a signal transducer, regulate key sperm phenotypes in at least two sperm compartments (summarized in *Figure 8*). First, in the sperm head, GIV's GEM activity is induced upon capacitation. Once activated, GIV modulates

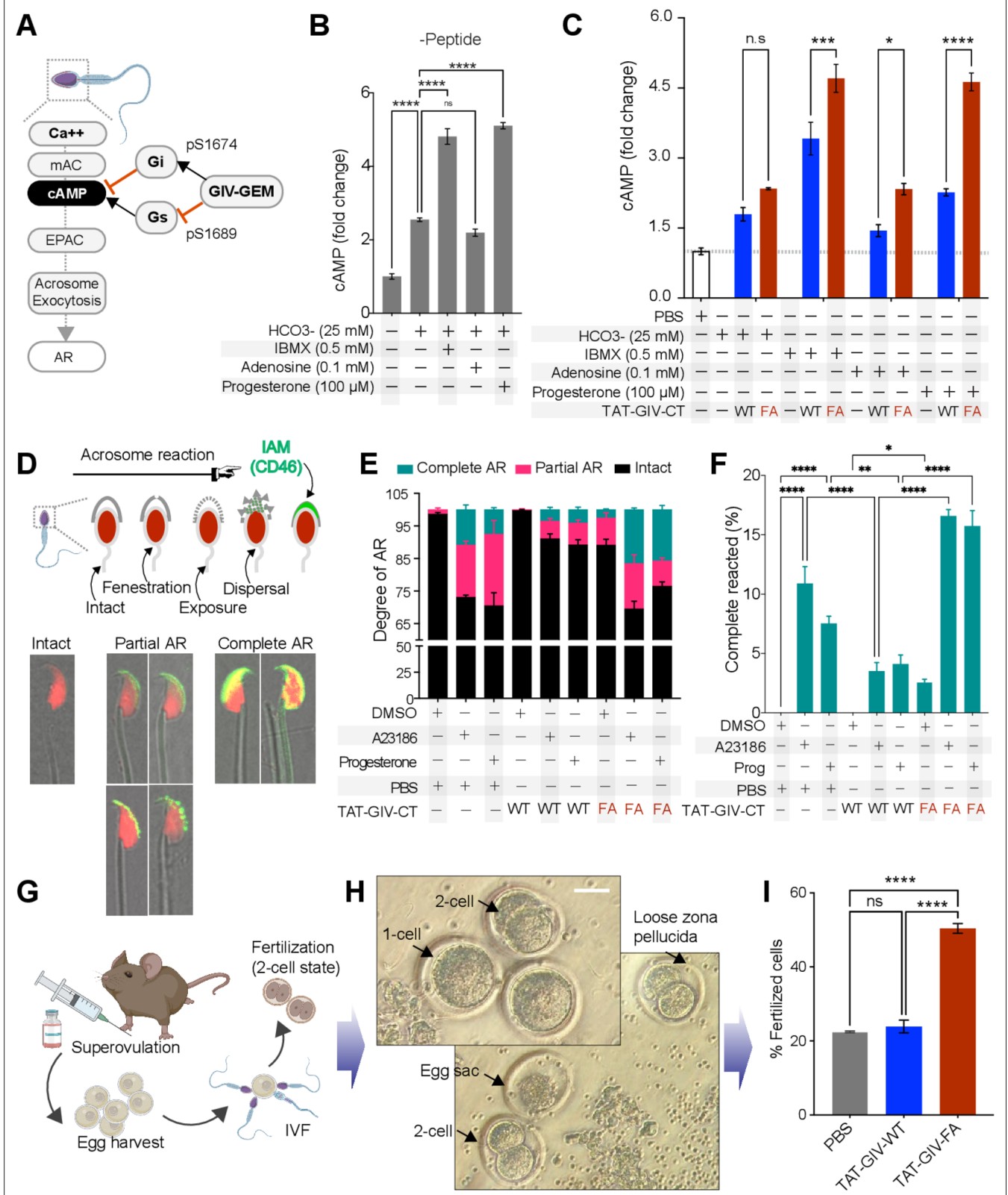

**Figure 7.** GIV's GEM function inhibits acrosomal reaction (AR). (**A**) Schematic summarizes the current knowledge of how Ca²⁺ and cAMP signaling regulates acrosome exocytosis during AR and how GIV's ability to modulate cAMP via both Gαi/s is hypothesized to impact AR. (**B**) Bar graphs display the fold change in cAMP in mouse sperms treated with various stimuli in the presence of DMSO. All results are presented as average ± SEM of three independent studies conducted on sperm isolated from three mice. Statistical significance was assessed using one-way analysis of variance (ANOVA)

*Figure 7 continued on next page*

*Figure 7 continued*

followed by Tukey's test for multiple comparisons. $^{ns}p > 0.05$, $****p < 0.0001$. (**C**) Bar graphs display the fold change in cAMP in TAT-GIVCT-transduced mouse sperms exposed to various stimuli. Dotted horizontal line represents cAMP concentration in PBS-treated samples, to which all other values were normalized. See also *Figure 7—figure supplement 1* for comparison of PBS vs. all other treatments and conditions with (+) or without (-) peptides. All results are presented as average ± SEM of three independent studies conducted on sperm isolated from three mice. Statistical significance was assessed using two-way ANOVA followed by Sidak's test for multiple comparisons. $*p < 0.05$, $***p < 0.001$, $****p < 0.0001$, $^{ns}p > 0.05$ (*Figure 7—source data 1*). (**D**) Schematic on top summarizes the assay used to quantify progressive changes in acrosome membrane during AR that was induced in vitro by exposing capacitated sperms to 10 µM A23186 or 100 µM progesterone. Images in the bottom panel are representative of acrosome-intact, partial AR and complete AR stages. (**E, F**) Stacked bar graphs in (**E**) display the proportion of sperms in each indicated condition that are either in partial or complete AR or with intact acrosomes. Bar graphs in (**F**) display just the relative proportion of sperms in (**E**) that have complete AR. All results are presented as average ± SEM of three independent studies conducted on sperm isolated from three mice. Statistical significance was assessed using one-way ANOVA followed by Tukey's test for multiple comparisons. $*p < 0.05$, $**p < 0.01$, $****p < 0.0001$ (*Figure 7—source data 2*). (**G–I**) Schematic in (**G**) displays the workflow used for in vitro fertilization (IVF) assays in (**H**, **I**). Representative images in (**H**) display the two -cell stage, which is quantified as % of total eggs in the assay and displayed as bar graphs in (**I**) as an indication of successful fertilization. Results are presented as average ± SEM of three independent studies conducted on sperm isolated from three mice. Statistical significance was assessed using one-way ANOVA including a Tukey's test for multiple comparisons. $****p < 0.0001$, $^{ns}p > 0.05$ (*Figure 7—source data 3*).

The online version of this article includes the following figure supplement(s) for figure 7:

**Source data 1.** Excel sheet with cAMP concentrations (corresponds to graph in *Figure 7B,C*).

**Source data 2.** Excel sheet with % cells with various stages of acrosomal reaction (AR) (corresponds to graph in *Figure 7D–F*).

**Source data 3.** Excel sheet with % fertilized cells (corresponds to graph in *Figure 7G–I*).

**Figure supplement 1.** An intact GEM motif in GIV is required for inhibiting cAMP surge and acrosomal reaction.

**Figure supplement 1—source data 1.** Excel sheet with cAMP concentrations (corresponds to graph in *Figure 7—figure supplement 1*).

---

both Gαi/s via the same GEM motif to suppress premature cAMP surges downstream of ligand-activated Gi/Gs-coupled GPCRs. Consequently, GIV-GEM inhibits premature acrosome shedding. Because both premature AR or failure to do so are important causes of male infertility (*Liu et al., 2006*), deciphering the signaling events that precisely regulate the timing of acrosome exocytosis has remained one of the most challenging and unresolved questions concerning mammalian reproductive biology (*Buffone et al., 2014*). Despite emerging evidence in the last decade that has challenged the long-held paradigms in the field, and mechanistic insights into sperm-extrinsic factors responsible for premature AR (*Sánchez-Cárdenas et al., 2021*; *Balestrini et al., 2021*; *Harper et al., 2004*), the identity of sperm-intrinsic pathways/processes/proteins that inhibit premature acrosome exocytosis was unknown. Our conclusion that GIV-GEM serves as a 'brake' for cAMP surge and prevents AR is consistent with the fact that the PDE-inhibitor sildenafil citrate (Viagra) increases cAMP to cause premature acrosomal reaction (*Glenn et al., 2007*). It is noteworthy that although canonical G protein signaling that is triggered by ligand-activated GPCRs has been implicated in the activation/inhibition of mACs and cAMP signaling in the sperm head (*Adeoya-Osiguwa et al., 2006*; *Schaefer et al., 1998*; *Flegel et al., 2016*), the role of non-canonical G protein we report here was never recognized previously. Because GIV is most highly expressed in sperm, the cAMP-regulatory role of GIV-GEM we define here implies that it may fulfill a major role in the regulation of cAMP in the sperm head. pYGIV was also detected in the sperm head, but its role in AR was not studied here. Because pYGIV activates class 1 PI3K, it is possible that the pYGIV→PI3K axis at that location could also influence rapid lipid phosphorylations that are also known to regulate AR (*Cohen et al., 2016*).

Second, in the sperm tail, GIV is an effector within the sAC→cAMP→PKA→multi-TK axis that gets robustly phosphorylated on Y1764; this site is known to directly bind and activate class 1 PI3Ks, which ultimately enhance Akt signals. GIV's GEM activity is also activated in the tails of capacitated sperms and enhances Akt signals, presumably via the previously defined GIV→Gi→'free' Gβγ→class 1 PI3K axis. These two mechanisms of Akt signaling have previously been shown to act as an 'AND' gate to maximally enhance Akt signaling in diverse cell types to increase cell survival and motility (*Lin et al., 2011*; *Lopez-Sanchez et al., 2015*) Furthermore, both GIV transcripts and its phosphoactivation by TKs (pYGIV) were reduced in sperms with lower motility. Because the global trend of progressive reduction in the number of motile and viable sperm in the ejaculate has been associated with a concomitant increase in the rates of infertility (*Dcunha et al., 2020*), our findings in the case of GIV add to the growing number of proteins that enrich the signal-ome of healthy sperm. For example, as an intrinsically disordered protein (IDP) and a multi-modular scaffold that generates crosstalk between

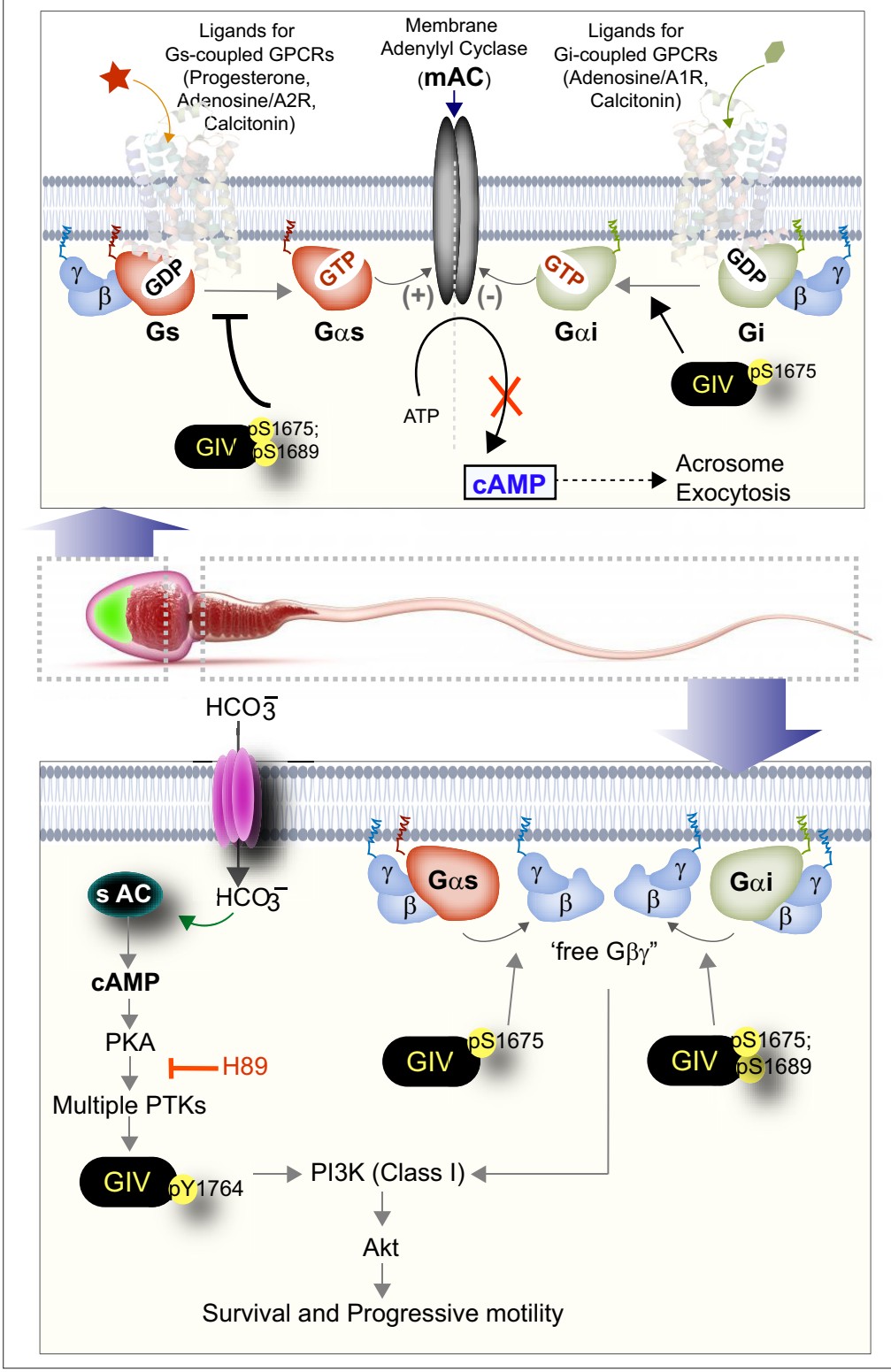

**Figure 8.** Summary and working model: spatiotemporally segregated roles of GIV/Girdin during sperm capacitation. Schematic summarizes the key findings in this work and places them in the context of existing literature. GIV is likely to primarily function during capacitation of sperm, during which it fulfills two key roles as a signal transducer in a spatiotemporally segregated manner. The first role (right, top) is in the head of the sperm, where GIV's GEM motif inhibits the AC→cAMP pathway and prevents acrosomal reaction. The second role (right, bottom) is in the mid-piece and tail region of the sperm, which involves tyrosine phosphorylation of GIV, which

*Figure 8 continued on next page*

*Figure 8 continued*

happens downstream of PKA activation. Such phosphorylation is rapidly induced during capacitation. In addition, GIV's GEM motif is activated and is required for the enhancement of PI3K/Akt signals, enhanced motility, and survival of sperms during capacitation.

diverse signaling pathways, GIV appears to be in a prominent position to orchestrate rapid cooperativity between these pathways and processes in the otherwise transcriptionally and translationally silent sperm cell.

In conclusion, our results provide evidence that GIV may perform different roles in the distinct spatial compartments of capacitating sperms. This study not only sheds light on defective GIV-signaling as potential 'marker' of male infertility, but also reveals that inhibitors of GIV-dependent signaling will inhibit fertility by reducing sperm motility and viability and by promoting premature AR. The latter is a promising strategy for the development of a male contraceptive 'pill' specifically targeting sperm.

# Materials and methods

## Key resources table

| Reagent type (species) or resource | Designation | Source or reference | Identifiers | Additional information |
|---|---|---|---|---|
| Antibody | Rabbit polyclonal anti-GIV (Girdin) (T-13) | Santa Cruz Biotechnology | sc-133371 | |
| Antibody | Rabbit monoclonal diagnostic grade anti-Girdin/GIV antibody | Custom; Sprint Bioscience | SP173 | Validated in prior publication *Ghosh et al., 2016* |
| Antibody | Rabbit polyclonal anti-GIV (Girdin) (CC-Ab) | Millipore Sigma | ABT80 | |
| Antibody | Rabbit polyclonal anti-GIV pS1675 Ab | Custom, from 21t Century Biosciences | n/a | Validated in prior publication *Bhandari et al., 2015* |
| Antibody | Rabbit polyclonal anti-GIV pS1689 Ab | Custom, from 21st Century Biosciences | n/a | Validated in prior publication *López-Sánchez et al., 2013* |
| Antibody | Rabbit monoclonal anti-GIV pY1764 Ab | Custom, Spring Biosciences Inc | n/a | Validated in prior publications *Midde et al., 2015*; *Lin et al., 2011*; *Midde et al., 2018*; *Dunkel et al., 2016* |
| Antibody | Rabbit monoclonal anti-pT308 AKT | Cell Signaling Technology | D9E | |
| Antibody | Mouse monoclonal anti-total AKT | Cell Signaling Technology | 40D4 | |
| Antibody | Mouse anti-sp56 | Thermo Fisher Scientific (Waltham, MA) | MA1-10866 | |
| Antibody | Mouse anti- human hexokinase 1/2 monoclonal antibody | R&D Systems, (Minneapolis, MN) | MAB8179 | |
| Antibody | Rabbit anti-phospho-PKA substrate (RRXS*/T*)100G7E | Cell Signaling Technology | 9624 | |
| Antibody | Goat anti-rabbit IgG, Alexa Fluor 594 conjugated | ThermoFisher Scientific | A11072 | For immunofluorescence (IF) |
| Antibody | Goat anti-mouse IgG, Alexa Fluor 488 conjugated | ThermoFisher Scientific | A11017 | For immunofluorescence (IF) |
| Antibody | IRDye 800CW goat anti-mouse IgG secondary (1:10,000) | LI-COR Biosciences | 926-32210 | For immunoblotting |
| Antibody | IRDye 680RD goat anti-rabbit IgG secondary (1:10,000) | LI-COR Biosciences | 926-68071 | For immunoblotting |

*Continued on next page*

*Continued*

| Reagent type (species) or resource | Designation | Source or reference | Identifiers | Additional information |
|---|---|---|---|---|
| Strain, strain background (*Mus musculus*) | $Ubc^{Cre-Ert2}$/+ x$Ccdc88a^{fl/fl}$ and $Ccdc88a^{fl/fl}$ mice | Masahide Takahashi (Nagoya University Graduate School of Medicine, Nagoya, Japan) | n/a | |
| Strain, strain background (*Mus musculus*) | WT and GIV-cKO (conditional KO) mice | This work | $Ubc^{Cre-Ert2}$/+ x$Ccdc88a^{fl/fl}$ (experimental group), and $Ubc^{Cre-Ert2/ Cre-Ert2}$ x $Ccdc88a^{fl/fl}$ (control group) | This work; male |
| Strain, strain background (*Mus musculus*) | C57BL/6 J mice (male and female) | The Jackson Laboratory | Stock number: 000664; Bar Harbor, ME | Male: source of sperm for biochemical, immunohistochemical, peptide transduction, and functional assays Female: for co-housing studies; source for eggs for IVF assays |
| Chemical compound | Paraformaldehyde 16% | Electron Microscopy Biosciences | 15710 | |
| Chemical compound | MTT | Millipore Sigma | 475989-1 GM | |
| Other | DAPI (4',6-Diamidino-2-Phenylindole, Dilactate) | Thermo Fisher Scientific | D3571 | Used in IF studies for staining DNA/nucleus |
| Kit/reagent | HisPur<sup>ä</sup> Cobalt Resin | Thermo Scientific | 89964 | |
| Kit/reagent | Glutathione Sepharose<sup>ä</sup> 4B | Sigma-Aldrich | GE17-0756-04 | |
| Chemical compound | Protease inhibitor cocktail | Roche | 11873580001 | |
| Chemical compound | Tyr phosphatase inhibitor cocktail | Sigma-Aldrich | P5726 | |
| Chemical compound | Ser/Thr phosphatase inhibitor cocktail | Sigma-Aldrich | P0044 | |
| Other | PVDF Transfer Membrane, 0.45 mM | Thermo Scientific | 88518 | Used for transfer in immunoblots |
| Commercial assay or kit | Countess II Automated Cell Counter | Thermo Fisher Scientific | AMQAX1000 | |
| Commercial assay or kit | Leica TCS SPE Confocal | Leica Microsystems | TCS SPE | |
| Commercial assay or kit | Light Microscope (brightfield images) | Carl Zeiss LLC | Axio Observer, Inverted; 491917-0001-000 | |
| Software | ImageJ | National Institute of Health | https://imagej.net/Welcome | |
| Software | Prism | GraphPad | https://www.graphpad.com/scientific-software/prism/ | |
| Software | LAS-X | Leica | https://www.leica-microsystems.com/products/microscope-software/p/leica-las-x-ls | |
| Software | Illustrator | Adobe | https://www.adobe.com/products/illustrator.html | |
| Software | ImageStudio Lite | LI-COR | https://www.licor.com/bio/image-studio-lite/ | |

Contact for reagent and resource sharing: Pradipta Ghosh (prghosh@ucsd.edu).

## Human subjects

Human sperm were collected from volunteers via masturbation, liquefied at room temperature for 30 min, and subsequently washed with TYH media before being exposed to non-capacitating (NC) TYH media or capacitating conditions (C) TYH plus 5 mg/ml of BSA and 15 mm $NaHCO_3$ for 4 hr at 37 °C and 5 % $CO_2$, as described previously (*Munné and Estop, 1993*). The study proposal was

approved by Institutional Review Board of University of California, San Diego (UCSD human subjects IRB protocol #16027, Gagneux). All samples were deidentified before use in studies. A written informed consent was obtained before participating in study. Consent to publish aggregate data with subject's anonymity was obtained. The study design and the use of human study participants were conducted in accordance to the criteria set by the Declaration of Helsinki.

## Mice

$Ubc^{Cre-Ert2/+}$ + $Ccdc88a^{fl/fl}$ and $Ccdc88a^{fl/fl}$ mice were generously provided by Dr. Masahide Takahashi (Nagoya University Graduate School of Medicine, Nagoya, Japan). Males $Ubc^{Cre-Ert2/+} \times Ccdc88a^{fl/fl}$ were bred to females $Ccdc88a^{fl/fl}$ to generate $Ubc^{Cre-Ert2/+} \times Ccdc88a^{fl/fl}$ (experimental group), and $Ubc^{Cre-Ert2/Cre-Ert2} \times Ccdc88a^{fl/fl}$ (control group) mice. Genotyping was performed by PCR, and only male mice were used in this study. Wildtype female C57BL/6 mice were purchased from The Jackson Laboratory (stock number: 000664; Bar Harbor, ME). All mice were housed in standard cages in an Association for Accreditation and Assessment of Laboratory Animal Care-approved animal facility at the University of California San Diego School of Medicine. This study was approved by the UCSD Institutional Animal Care and Use Committee (protocol #S17223; Ghosh), which serves to ensure that all federal guidelines concerning animal experimentation are met.

For all biochemical, immunofluorescence, peptide transduction, and functional studies, the source of sperms was C57BL/6 J mice, which were bred and housed under another protocol, which was also approved by the UCSD Institutional Animal Care and Use Committee (protocol #S16223; Gagneux).

## Reagents and antibodies

All reagents were of analytical grade and obtained from Sigma-Aldrich (St. Louis, MO) unless otherwise stated.

The affinity-purified anti-pS1689-GIV and pS1674-GIV were generated commercially in collaboration with 21st Century Biochemicals (Marlboro, MA) and validated previously (**López-Sánchez et al., 2013**; **Bhandari et al., 2015**.) Rabbit anti-GIV CT (T-13) was obtained from Santa Cruz Biotechnology; and a previously validated custom-raised anti-p-GIV (pY1764) was from Spring Bioscience (Pleasanton, CA, USA) (**Midde et al., 2015**; **Lin et al., 2011**; **Midde et al., 2018**; **Dunkel et al., 2016**). Mouse mAbs against pTyr was from BD Biosciences; mouse anti-His, anti-a tubulin and anti-actin were obtained from Sigma; rabbit monoclonal anti-phospho-(p)Akt (Thr308) and anti-total (t)Akt were from Cell Signaling Technology (Beverly, MA). Rabbit anti-GIV-coiled-coil (CC) was obtained from EMD Millipore (Carlsbad, CA). Other commercially obtained antibodies used in this work were mouse anti-sp56 (MA1-10866) was purchased from Thermo Fisher Scientific (Waltham, MA), mouse anti-human hexokinase 1/2 monoclonal antibody (catalog # MAB8179; R&D Systems, Minneapolis, MN), and rabbit anti-phospho-PKA substrate (RRXS*/T*) (100G7E) mAb #9624 from Cell Signaling Technology.

Goat anti-rabbit and goat anti-mouse Alexa Fluor 680 or IRDye 800 F (ab0)2 used for immunoblotting were from LI-COR Biosciences (Lincoln, NE). Goat anti-rabbit Alexa Fluor 488 and goat anti-mouse Alexa Fluor 594 for immunofluorescence were purchased from Life Technologies.

## IHC of mouse testes

Mouse testes were fixed in zinc paraformaldehyde to prepare FFPE tissue blocks. Tissue sections of 4 μm thickness were cut and placed on glass slides coated with poly-L-lysine, followed by deparaffinization and hydration. Heat-induced epitope retrieval was performed using sodium citrate buffer (pH 6) in a pressure cooker. Tissue sections were incubated with 3% hydrogen peroxidase for 10 min to block endogenous peroxidase activity, followed by incubation with primary antibody overnight in a humidified chamber at 4 °C. Antibodies used for immunostaining were SP173 rabbit monoclonal, anti-GIV antibody. Immunostaining was visualized with a labeled streptavidin–biotin using 3,3'-diaminobenzidine as a chromogen and counterstained with hematoxylin.

## Source of live mouse and human sperms

Mouse sperm suspension were obtained from cauda epididymis of mature male (9 weeks old) placed in 1 ml NC buffer prewarmed at 38.1 °C for 25 min in siliconized/low-adhesion microfuge tubes. The caudae epididymis was cut to let the spermatozoa swim out. The microfuge tube was agitated on an orbital shaker for 10 min to facilitate the swim out of the sperm. The tubes were then placed upright

on a bench top, and the epididymal tissues were allowed to settle for 10 min. The sperm suspension was then removed from the top, and an aliquot was taken to ensure purity and for counting (~99%).

Human ejaculates were collected from volunteers via masturbation after 1 week of abstinence under UCSD human subject protocol (UCSD human subjects IRB protocol #16027, Gagneux). After liquefaction at room temperature, 1 ml ejaculate was transferred into the bottom of 1 ml prewarmed NC buffer and incubated for additional 1 hr to swim up procedure. Highly motile sperm mobilized to the upper layer was collected for the experiment.

### In vitro capacitation and induction of the AR

Freshly obtained human and mouse sperms were segregated into low-motile and high-motile populations using 'swim-up' technique. Subsequently highly motile sperms were capacitated in TYH buffer containing 5 mg/ml BSA and 15 mM $NaHCO_3$ at 37 °C under 5 % $CO_2$ for the indicated time mentioned in figure legends. Sperms were both lysed in reducing sample buffer for immunoblotting and fixed in 3% paraformaldehyde for immunofluorescence staining. Acrosomal reaction in sperm was triggered by incubating capacitated sperm either with Ca2+ ionophore or progesterone for the indicated time at 37 °C in 5 % $CO_2$. Sperm were then fixed and co-stained for peanut agglutinin (PNA-488; green, an acrosomal marker) and either pYGIV, or pSerGIV and DAPI.

### Confocal immunofluorescence

Sperms were fixed with 3 % paraformaldehyde in PBS for 25 min at room temperature, treated with 0.1 M glycine for 10 min, and subsequently blocked/permeabilized with PBS containing 1 % BSA and 0.1 % Triton X-100 for 20 min at room temperature. Primary and secondary antibodies were incubated for 1 hr at room temperature in PBS containing 1 % BSA and 0.1 % Triton X-100. Dilutions of antibodies used were as follows: GIV (1:500); phospho-GIV (Tyr1764; 1:500); phospho-pan-Tyr (1:500); α-tubulin (1:500); phosphor-GIV (Ser1674; 1:500); phosphor-GIV (Ser1689; 1:500); Peanut agglutinin (PNA) (1:500); His (1:500) and DAPI (1:2000). Secondary Alexa conjugated antibodies were used at 1:500 dilutions.

In the case of frozen sections of mouse testes, the protocol used was as follows: cryosections were washed three times with PBS, followed by 0.15 % glycine for 10 min at room temperature and incubated for 20 min in blocking buffer (1 % BSA in PBS), then 2 hr in primary antibodies and 45 min in secondary antibodies. Dilutions of antibodies used were as follows: GIV (1:500); phospho-GIV (Tyr1764; 1:250); ZP3R (1:500); DAPI (1:1000). Secondary Alexa conjugated antibodies were used at 1:250 dilutions. Sperms and sections were imaged on a Leica SPE confocal microscope using a 63× oil objective using 488, 561, 633, and 405 laser lines for excitation. The settings were optimized, and the final images scanned with line-averaging of 3. All images were processed using ImageJ software (NIH) and assembled for presentation using Photoshop and Illustrator software (Adobe).

### Dual-color quantitative immunoblotting

Protein samples were separated by SDS/PAGE and transferred to PVDF membranes (Millipore). Membranes were blocked with PBS supplemented with 5 % nonfat milk (or with 5 % BSA when probing for phosphorylated proteins) before incubation with primary antibodies. Infrared imaging with two-color detection and quantification were performed using a LI-COR Odyssey imaging system. Primary antibodies were diluted as follows: anti-His 1:1000; anti-GIV (tGIV) 1:500; anti-phospho-Tyr-1764-GIV (pYGIV) 1:500; anti-phospho-Tyr (pan pY) 1: 500; anti-phospho-PKA 1: 500; anti-hexokinase 1:1000; anti-phospho-Akt (Thr308) 1:500; anti-Akt 1:500; anti-β-tubulin 1:1000. All Odyssey images were processed using ImageJ software (NIH) and assembled for presentation using Photoshop and Illustrator software (Adobe).

### His-TAT purification and transduction in sperms

Cloning of TAT-GIV-CT-WT and TAT-GIV-CT-FA mutant has been described (*Ma et al., 2015*). TAT-constructs were expressed using BL21(DE3)-pLysS (Invitrogen) and Terrific Broth (BioPioneer) supplemented with additives as per auto-induction protocols outlined by *Studier, 2005*. Briefly, cultures of bacteria were grown at 300 rpm at 37 °C for 5 hr, then at 25 °C overnight. Cells were lysed in 10 ml of lysis buffer containing 20 mM Tris, 10 mM imidazole, 400 mM NaCl, 1 % (vol:vol) sarkosyl, 1 % (vol:vol) Triton X-100, 2 mM DTT, 2 mM Na3oV4 and protease inhibitor mixture (Roche Diagnostics), pH 7.4,

sonicated (3 × 30 s), cleared at 12,000× g for 20 min at 4 °C and affinity-purified on Ni-NTA agarose resin (Qiagen) (4 hr at 4 °C). Proteins were eluted in elution buffer containing 20 mM Tris, 300 mM imidazole, 400 mM NaCl, pH 7.4, dialyzed overnight against TBS containing 400 mM NaCl and stored at −80 °C.

TAT transduction in sperms was performed by incubating them with 400–800 nM TAT-GIV-CT peptides for 30 min. Efficiency uptake was measured by flow cytometry. Different batches of TAT-GIV-CT protein preparations were used for optimization of equal uptake of recombinant TAT-GIV-CT WT and FA peptides. Optimization steps included timing of transduction, wash step, and concentrations of each peptide used to ensure that WT and FA peptides are equal in sperm. The recombinant protein that showed the most efficient uptake was subsequently used in four different mouse sperm samples to document consistent uptake test by western blotting, by FACS and immunofluorescence, and finally, to confirm that GIV peptides retain functionality (G protein binding) upon uptake.

## CASA system

Freshly obtained sperms were segregated into low-motile and high-motile populations using 'swim-up' technique and highly motile sperms were subsequently capacitated in TYH buffer containing 5 mg/ml BSA and 15 mM $NaHCO_3$ along with TAT-GIV-CT peptides at 37 °C under 5 % $CO_2$ for 3 hr. The sperm motility and progressive motility were measured on CASA on a Hamilton Thorne IVSO-CASA (Berns Laboratory, UC San Diego).

## Measurement of sperm cAMP level

Mouse sperms at a density of $2 \times 10^7$ cells/ml ($6 \times 10^6$ cells in total) were first peptide transduced with TAT-GIV-CT for 30 min, washed gently with PBS three times to remove excess peptides before their use in cAMP assays. Peptide-transduced sperms were pre-incubated with 0.5 mM isobutyl methyl xanthine (IBMX) prior to exposure to various chemicals at the following final concentrations: 25 mM $NaHCO_3$, 0.1 mM adenosine, or 100 µM. After mixing with the respective stimulus, the samples were incubated for 30 min at 37 °C, followed by the addition of 0.25 M HCl (final concentration) to quench the biochemical reactions. After incubation for 30 min at room temperature, cell debris was sedimented by centrifugation at 3000 g for 5 min at room temperature. The cAMP concentration in the supernatant was determined by a competitive enzyme immunoassay according to the product manual (catalog # ADI-900-066, Enzo Life Sciences).

## Tamoxifen treatment and natural mating

4–5- week-old mice experimental or control mice received an intraperitoneal (i.p.) tamoxifen injection 1 mg/100 µl/mouse/day (Millipore Sigma, St. Louis, MO) solubilized in 100 % corn oil for five consecutive days. *Ccdc88a* gene knockout (GIV-knockout) was confirmed using DNA qPCR as described in our previous study (*Swanson et al., 2020*). Mice were then housed for 3 weeks with random females to promote mating with the intent to discharge sperms in which tamoxifen has not yet induced Cre expression.

After 3 weeks, each male mouse was housed with three 7–8- week-old C57BL/6 fertile females for 3 months. The frequency of successful live births and the litter size was recorded. After 3 months, experimental and control males were sacrificed, and testicles and epididymis were collected for IHC, immunoblotting, and mRNA analysis.

## Transcriptomic datasets from infertile patients

Publicly available microarray (GSE4797 *Feig et al., 2007*, E-TABM-234 *Feig, 2008*, GSE6872 *Platts et al., 2007*, GSE26881 *Pacheco et al., 2011*) and RNASeq dataset (GSE103905 *Winge et al., 2018*) were downloaded from the National Center for Biotechnology Information (NCBI) Gene Expression Omnibus website (GEO) and ArrayExpress. The data was processed using the Hegemon data analysis framework repositories for male infertility (*Sahoo, 2012*; *Sahoo et al., 2008*; *Sahoo et al., 2010*). Microarray datasets (GSE4797; GSE6872; GSE26881; E-TMAB-234) were normalized using robust multi-array average (RMA). RNA-sequencing dataset (GSE103905) was normalized using transcripts per millions (TPM) normalization method; for downstream analyses, log2(TPM) if TPM >1 and (TPM – 1) if TPM <1 is used. Distribution of gene expression values is illustrated using boxplots and mean as circle with 95 % confidence intervals as arrows. Numbers on top indicate the p values, which were

derived from Welch's t-test. All semen samples mentioned in the above datasets were classified based on WHO (2010) guidelines for semen parameters (*Cooper et al., 2010*).

| WHO (2010) guidelines Cooper et al., 2010 (Semen parameters) | Motility (%) | Morphology (%) | Concentration ($10^6$/ml) |
|---|---|---|---|
| Fertile individual | ≥40 | ≥4 | ≥15 |

## Total RNA isolation

Total RNA from mouse testicles were isolated using the Direct-zol RNA Miniprep Kit (Zymo Research, Irvine, CA) following the manufacturer's instructions. RNA concentration was measured using the NanoDrop One (ThermoScientific, Waltham, MA). The RNA Integrity Number (RIN) was assessed using the 4200 TapeStation system and the TapeStation RNA ScreenTape & Reagents (Agilent Technologies, Santa Clara, CA).

## RNA-seq and data analysis

Total RNA samples were submitted to the IGM Genomics Center (University of California San Diego) for library preparation and sequencing. mRNA stranded sequencing libraries were generated with the TruSeq Stranded mRNA Sample Prep Kit with TruSeq Unique Dual Indexes (Illumina, San Diego, CA). Resulting libraries were multiplexed and sequenced with 100 bp paired-end reads (PE100) to a depth of approximately 30 million reads per sample on an Illumina NovaSeq 6000. Samples were demultiplexed using bcl2fastq v2.20 Conversion Software (Illumina).

To determine which genes were differentially expressed in WT and GIV-knockout mice testes, transcript-level abundance of paired-end RNA-seq data was estimated by Salmon (1.1.0) using the mouse transcriptome from Genecode (vM24). Tximport (1.14.2) was used to aggregate transcript-level quantification to the gene level. The resulting gene counts were used as an input to DESeq2 Bioconductor package. Differentially expressed genes below a Benjamini-Hochberg (BH)-adjusted p-value of 0.05 were considered significant. Also, all genes differentially expressed were included in REACTOME pathway enrichment analysis. Statistically significant pathways of upregulated and downregulated DEGs are listed in the table and bar plots of upregulated and downregulated enriched pathways. In Gene Set Enrichment Analysis (GSEA) analysis, some fertility-related gene sets from Molecular Signatures Database (MSigDB) were tested. Genes from these gene sets were used to rank order the samples and test for GIV-knockout versus WT phenotype classification using the area under the curve (AUC) receiver operating characteristics (ROC) curve and displayed such classification using violin plots.

## Statistical analysis

Statistical significance between datasets with three or more experimental groups was determined using one-way analysis of variance (ANOVA) followed by Tukey's test for multiple comparisons. Unpaired t-test is used to test the statistical difference between two experimental groups. For all tests, a p-value>0.05 is considered as significant. All experiments were repeated at least three times. All statistical analyses were performed using GraphPad Prism 9.

## Acknowledgements

We thank Masahide Takahashi, Ph.D (Nagoya University, Japan) for sharing Ccdc88afl/fl-Ubc-Cre-Ert2/+ mice and Lee Swanson for assisting with the initial breeding of the colonies. We thank members in the laboratory of Pamela L Mellon, Ph.D. (UCSD) for helpful technical suggestions along the way. We thank members of the Michael Berns laboratory for help with the CASA machine. This work was supported by the National Institute of Health Grants: CA238042, CA100768, AI141630 and CA160911 (to Pr.Gh.), GM095882 (to Pa.Ga) and GM138385 (to DS). GDK was supported through The American Association of Immunologists Intersect Fellowship Program for Computational Scientists and Immunologists. IL-S by the American Heart Association (AHA #14POST20050025) C.R was supported, in part, by an NIH-funded Training Grant Programs (T32 DK007202, T32 CA121938). This publication includes data generated at the UC San Diego IGM Genomics Center Utilizing an Illumina

NOVASeq 6000 that was purchased with funding from a National Institutes of Health SIG grant (#S10 OD026929).

## Additional information

### Funding

| Funder | Grant reference number | Author |
|---|---|---|
| National Cancer Institute | CA100768 | Pradipta Ghosh |
| National Cancer Institute | CA238042 | Pradipta Ghosh |
| National Institute of Allergy and Infectious Diseases | AI141630 | Pradipta Ghosh |
| National Institute of General Medical Sciences | GM095882 | Pascal Gagneux |
| National Cancer Institute | CA160911 | Pradipta Ghosh |
| American Association of Immunologists | Intersect fellowship | Gajanan Dattatray Katkar |
| National Institute of Diabetes and Digestive and Kidney Diseases | T32 DK007202 | Cristina Rohena |
| National Cancer Institute | T32 CA121938 | Cristina Rohena |
| National Institute of General Medical Sciences | GM138385 | Debashis Sahoo |
| American Heart Association | 14POST20050025 | Inmaculada Lopez-Sanchez |

The funders had no role in study design, data collection and interpretation, or the decision to submit the work for publication.

### Author contributions

Sequoyah Reynoso, Sahar Taheri, Data curation, Formal analysis; Vanessa Castillo, Data curation, Methodology; Gajanan Dattatray Katkar, Data curation, Formal analysis, Methodology, Visualization, Writing – review and editing; Inmaculada Lopez-Sanchez, Celia Espinoza, Cristina Rohena, Data curation, Formal analysis, Methodology; Debashis Sahoo, Data curation, Formal analysis, Funding acquisition, Methodology; Pascal Gagneux, Data curation, Formal analysis, Funding acquisition, Methodology, Resources, Validation, Visualization, Writing – review and editing; Pradipta Ghosh, Conceptualization, Formal analysis, Funding acquisition, Investigation, Project administration, Resources, Supervision, Validation, Visualization, Writing – original draft, Writing – review and editing

### Author ORCIDs

Vanessa Castillo http://orcid.org/0000-0002-4182-8846
Pascal Gagneux http://orcid.org/0000-0001-9599-9838
Pradipta Ghosh http://orcid.org/0000-0002-8917-3201

### Ethics

Human sperm were collected from volunteers via masturbation. The study proposal was approved by Institutional Review Board of University of California, San Diego (UCSD human subjects IRB protocol #16027). All samples were deidentified before use in studies. A written informed consent was obtained before participating in study. Consent to publish aggregate data with subject's anonymity was obtained. The study design and the use of human study participants was conducted in accordance to the criteria set by the Declaration of Helsinki.

All mice were housed in standard cages in an Association for Accreditation and Assessment of Laboratory Animal Care-approved animal facility at the University of California San Diego School of Medicine. This study was approved by the UCSD Institutional Animal Care and Use Committee (Protocol #S17223; Ghosh), which serves to ensure that all federal guidelines concerning animal experimentation

are met. For all biochemical, immunofluorescence, peptide transduction and functional studies, the source of sperms was C57BL/6J mice, which were bred and housed under another protocol, which was also approved by the UCSD Institutional Animal Care and Use Committee (Protocol #S16223; Pascal).

## Decision letter and Author response

Decision letter https://doi.org/10.7554/eLife.69160.sa1
Author response https://doi.org/10.7554/eLife.69160.sa2

## Additional files

### Supplementary files
• Transparent reporting form

### Data availability

Sequencing data have been deposited in GEO under accession codes GSE171704.

The following dataset was generated:

| Author(s) | Year | Dataset title | Dataset URL | Database and Identifier |
|---|---|---|---|---|
| Sequoyah R | 2021 | GIV/Girdin Regulates Spatiotemporal Signaling during Sperm Capacitation and is Required for Male Fertility | https://www.ncbi.nlm.nih.gov/geo/query/acc.cgi?acc=GSE171704 | NCBI Gene Expression Omnibus, GSE171704 |

The following previously published datasets were used:

| Author(s) | Year | Dataset title | Dataset URL | Database and Identifier |
|---|---|---|---|---|
| Feig C | 2007 | Microarray analysis of human spermatogenic dysfunction | https://www.ncbi.nlm.nih.gov/geo/query/acc.cgi?acc=GSE4797 | NCBI Gene Expression Omnibus, GSE4797 |
| Feig C | 2008 | Transcription profiling of human testis samples from men with highly defined and homogenous testicular pathologies reveals patterns that correlate with distinct stages of spermatogenesis | https://www.ebi.ac.uk/arrayexpress/experiments/E-TABM-234/ | ArrayExpress, E-TABM-234 |
| Platts AE | 2007 | Spermatozoal RNA Profiles (U133 Plus 2.0 Array) | https://www.ncbi.nlm.nih.gov/geo/query/acc.cgi/GSE6872 | NCBI Gene Expression Omnibus, GSE6872 |
| Winge SB | 2018 | Transcriptome analysis of adult Klinefelter testis tissue samples compared to controls | https://www.ncbi.nlm.nih.gov/geo/query/acc.cgi/GSE103905 | NCBI Gene Expression Omnibus, GSE103905 |
| Pacheco SE | 2011 | mRNA Content of Human Sperm | https://www.ncbi.nlm.nih.gov/geo/query/acc.cgi/GSE26881 | NCBI Gene Expression Omnibus, GSE26881 |

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
