## [Decision Letter]

**Acceptance summary:**

This work is of interest to the field of reproduction. Prior to fertilization, spermatozoa undergo a series of morphological and biochemical changes to become fertilization competent, driven by a rapid and poorly understood signaling cascade, culminating in the acrosome reaction. This latter reaction releases to the outside components from a vesicle, the acrosome, in the spermatozoan head and transforms the head plasma membrane so that sperm can fuse with the egg. The work shows that a G protein modulator GIV/Girdin, influences sperm motility and the acrosome reaction. In so doing it is important for fertilization and is one more strategy to control untimely acrosome reaction. The proposed mechanism is well supported by a variety of different experimental approaches.

**Decision letter after peer review:**

Thank you for submitting your article "GIV/Girdin, a Non-receptor Modulator for Gαi/s, Regulates Spatiotemporal Signaling during Sperm Capacitation and is Required for Male Fertility" for consideration by *eLife*. Your article has been reviewed by 2 peer reviewers, and the evaluation has been overseen by Jonathan Cooper as the Senior and Reviewing Editor. The reviewers have opted to remain anonymous.

Essential revisions:

As you will see from the full reviews below, the comments should not require new experiments. However, you should fully address the concerns with addition of quantification and rewriting. Please pay particular attention to the comments in the "Recommendations for Authors" from all reviewers regarding quantification and statistics, and be sure to include relevant literature citations where noted.

*Reviewer #1 (Recommendations for the authors):*

The authors indicate the regulatory role of GIV is the first mechanism reported to control untimely acrosome reaction which is incorrect. On the other hand, there are several aspects of the presented results, many of which are related to quantitation, that require improvement before the paper can be published. They are listed in what follows:

1) It has been reported that H89 is not very specific and it can even inhibit CatSper. How do the authors discard the possibility that it is this channel or intracellular Ca^2+^ that are being modulated by GIV?

2) In line 190 there is a point missing.

3) It is impossible to evaluate quantitative immunolocalization assertions from looking at one or two sperm. In this regard the work needs better quantitation. In the legend, the authors indicate their images are representative and do not even bother to indicate of how many independent mice or human samples were used and how many cells were examined. The results should include box plots with the summary of results from all sperm examined.

4) There are really no quantitative results regarding acrosome reaction as only one or two cells are shown.

5) Line 233; it seems it should be suggest, not suggests.

6) Line 244. Again the quantitation is superficial, how many experiments, where are the numbers?

7) Are the Y axis wrongly labeled in Figure 6G. Where is the percentage of hyperactivated motility? Why are the values so low? Normally hyperactivation after capacitation is 15 or 20 %.

8) Line 262. This statement is quite incomplete and partial. Where are the references that have indicated the participation of mAC in several systems, from sea urchin, mouse and human sperm? For example:

Spehr M, Schwane K, Riffell JA, Barbour J, Zimmer RK, Neuhaus EM, Hatt H.

Particulate adenylate cyclase plays a key role in human sperm olfactory

receptor-mediated chemotaxis. J Biol Chem. 2004 Sep 17;279(38):40194-203. doi:10.1074/jbc.M403913200. Epub 2004 Jul 22. PMID: 15271985.

Baxendale RW, Fraser LR. Evidence for multiple distinctly localized adenylyl cyclase isoforms in mammalian spermatozoa. Mol Reprod Dev. 2003 Oct;66(2):181-9. doi: 10.1002/mrd.10344. PMID: 12950106.

Beltrán C, Vacquier VD, Moy G, Chen Y, Buck J, Levin LR, Darszon A.

Particulate and soluble adenylyl cyclases participate in the sperm acrosomereaction. Biochem Biophys Res Commun. 2007 Jul 13;358(4):1128-35. doi:10.1016/j.

9) Figure 7. The authors never indicate the number of independent experiments performed and they should. I guess all experiments in Figure 7 are done with capacitated sperm which should be indicated. Progesterone does not trigger AR in non-capacitated sperm.

10) In Figure 7C. Why does bicarbonate not stimulate but actually decreases cAMP when it stimulates sAC without peptide, as shown in B? Why does WT peptide stimulate above PBS, it should decrease cAMP?

11) In line 277 there is a space between comas that should be deleted.

12) In Figure 7 why does A23187 induce such a low % of full AR.

13) The authors are ignoring work that has indicated mechanisms that control premature acrosome reaction such as:

Harper, C. V., Barratt, C.L.R.R., Publicover, S.J., 2004. Stimulation of human spermatozoa with progesterone gradients to simulate approach to the oocyte. Induction of [Ca^2+^]i oscillations and cyclical transitions in flagellar beating. J. Biol. Chem. 279, 46315-46325. https://doi.org/10.1074/jbc.M401194200

Balestrini PA, Sanchez-Cardenas C, Luque GM, Baro Graf C, Sierra JM,

Hernández-Cruz A, Visconti PE, Krapf D, Darszon A, Buffone MG. Membrane hyperpolarization abolishes calcium oscillations that prevent induced acrosomal exocytosis in human sperm. FASEB J. 2021 Jun;35(6):e21478. doi:10.1096/fj.202002333RR. PMID: 33991146.

Sánchez-Cárdenas C, Romarowski A, Orta G, De la Vega-Beltrán JL, Martín-Hidalgo D, Hernández-Cruz A, Visconti PE, Darszon A. Starvation induces an increase in intracellular calcium and potentiates the progesterone-induced mouse sperm acrosome reaction. FASEB J. 2021 Apr;35(4):e21528. doi:10.1096/fj.202100122R

14) It is unfair not to indicate that the group of Mayorga in Argentina has described since some years the fundamental role of small G proteins like Rab in the acrosome reaction.

15) Were mouse epidydimal sperm collected by swim out or swim up, please indicate?

16) Please indicate the progesterone and A23187 concentrations used.

17) Line 414 should be were not where.

18) In human sperm capacitation for 3 hours is barely enough.

*Reviewer #3 (Recommendations for the authors):*

It is not clear how to interpret the absence of GIV transcripts from "Sertoli Cell Only" testes. These testes lack all germ cells. Absence of GIV confirms that GIV is expressed in male germ cells but says little about the role of GIV in fertility or sperm function. It may make sense to recalculate the significance in Figure 2E, excluding the Sertoli cell only samples to focus on samples from men who make defective sperm.

The word "sperm" appears in the Impact Statement and should be replaced with "spermatozoa".

---

## [Author Response]

Essential revisions:As you will see from the full reviews below, the comments should not require new experiments. However, you should fully address the concerns with addition of quantification and rewriting. Please pay particular attention to the comments in the "Recommendations for Authors" from all reviewers regarding quantification and statistics, and be sure to include relevant literature citations where noted.

We thank the Editor for this overview of what we must address to improve the manuscript. We have outlined below exactly how each quantification/re-analyses changed (or did not change) the data and how the text was modified to address/mitigate the concerns raised.

Reviewer #1 (Recommendations for the authors):The authors indicate the regulatory role of GIV is the first mechanism reported to control untimely acrosome reaction which is incorrect. On the other hand, there are several aspects of the presented results, many of which are related to quantitation, that require improvement before the paper can be published.

We apologize for this unfortunate impression; this was never our intension. We have meticulously inserted citations recommended by this reviewer to ensure that due credit is given to prior work. As for the numerous issues pointed out in the results and display items (concerning missing information regarding quantification), we have thoroughly edited the legends, methods and Results sections in each instance to include the necessary information. We hope that the edits effectively mitigate this reviewer’s concerns. We are also grateful to this reviewer for his/her time and effort that went into providing an in-depth review that picked up unintended error and typos.

They are listed in what follows:1) It has been reported that H89 is not very specific and it can even inhibit CatSper. How do the authors discard the posibility that it is this channel or intracellular Ca^2+^ that are being modulated by GIV?

The reviewer is right in that, we cannot rule out that in the studies where we used H89 to inhibit PKA, that we did not inadvertently also inhibit Ca channels such as Catsper.

Action(s) taken: We have inserted two relevant citations and discussed this possibility in “results and discussion” right after we draw conclusions from the H89 studies in Figure 3. For the convenience of the Editor/Reviewers, we have copied and pasted the sentence below:

“Because the sperm Ca^2+^ channel, Catsper exerts both spatial and temporal control over tyrosine phosphorylation as sperm acquire the capacity to fertilize^32^, and there is some evidence that H89 may directly inhibit Catsper^33^, the contributions of a possible alternative Ca^2+^→TK→pYGIV pathway towards sperm motility cannot be ruled out.”.

2) In line 190 there is a point missing.

We have corrected this error.

3) It is impossible to evaluate quantitative immunolocalization assertions from looking at one or two sperm. In this regard the work needs better quantitation. In the legend, the authors indicate their images are representative and do not even bother to indicate of how many independent mice or human samples were used and how many cells were examined. The results should include box plots with the summary of results from all sperm examined.

We agree that a montage of sperm images with spatially segregated phosphoevents, in the absence of some form of quantitative analyses is impossible to assess for rigor and reproducibility.

Action(s) taken: Because spatial patterns of staining is what we saw changing and hence, we wanted to claim, we have now provided specifics of how many times each study was carried out and in how many mice, and how many sperms were evaluated before drawing a conclusion. In doing so, we hope that we have now adequately defined the term ‘representative’ and conveyed rigor. These details are expanded in each figure legend where IF images are displayed.

4) There are really no quantitative results regarding acrosome reaction as only one or two cells are shown.

In the revised version of the manuscript, we have now provided specifics of how many times each study was carried out and in how many mice, and how many sperms were evaluated before drawing a conclusion. In doing so, we hope that we have now adequately defined the term ‘representative’ and conveyed rigor. These details are expanded in each figure legend where IF images are displayed. Because no quantitative claims were made, staining was never quantified.

5) Line 233; it seems it should be suggest, not suggests.

We have replaced ‘suggests’ with ‘suggest’.

6) Line 244. Again the quantitation is superficial, how many experiments, where are the numbers?

This statement refers to Figure 6 (TAT-GIV transduction studies). We have now included details in figure legends for how many repeats were performed. For the convenience of the Editor/Reviewer, here are the details, which we have now included in Methods and Figure 6 legend:

1. 3-4 different batches of protein preps were used for optimization of balanced uptake of TAT-GIV-CT WT and FA peptides. Optimization steps included timing of transduction, wash step, and concentrations of each peptide used.

2. The recombinant protein that showed the most efficient uptake was subsequently used in *four different mouse sperm samples* to document consistent uptake test by western blotting, by FACS and immunofluorescence, and finally, to confirm that GIV peptides retain functionality (G protein binding) upon uptake.

3. For panels 6D-I, the graphed findings were reproducibly observed in sperm samples from *3 different mice*, conducted on 3 different days.

7) Are the Y axis wrongly labeled in Figure 6G. Where is the percentage of hyperactivated motility? Why are the values so low? Normally hyperactivation after capacitation is 15 or 20 %.

There is no error in the Y axis labeling. The lower than expected % hypermotlity (progressive) that we observe in this assay is consistent with the fact that CASA was not done immediately after sperm isolation. Instead, there was a delay due to TAT-peptide transduction related steps (or control incubation in PBS). As stated in methods, our analyses could only be done at 3h. It is entirely possible that the reduce % hypermotility is due to this delay, as has been shown by others^1^.

8) Line 262. This statement is quite incomplete and partial. Where are the references that have indicated the participation of mAC in several systems, from sea urchin, mouse and human sperm? For example:Spehr M, Schwane K, Riffell JA, Barbour J, Zimmer RK, Neuhaus EM, Hatt H.Particulate adenylate cyclase plays a key role in human sperm olfactoryreceptor-mediated chemotaxis. J Biol Chem. 2004 Sep 17;279(38):40194-203. doi:10.1074/jbc.M403913200. Epub 2004 Jul 22. PMID: 15271985.Baxendale RW, Fraser LR. Evidence for multiple distinctly localized adenylyl cyclase isoforms in mammalian spermatozoa. Mol Reprod Dev. 2003 Oct;66(2):181-9. doi: 10.1002/mrd.10344. PMID: 12950106.Beltrán C, Vacquier VD, Moy G, Chen Y, Buck J, Levin LR, Darszon A.Particulate and soluble adenylyl cyclases participate in the sperm acrosomereaction. Biochem Biophys Res Commun. 2007 Jul 13;358(4):1128-35. doi:10.1016/j.

On Page 12 of this revised submission, we have added these references and expanded on the conserved role of mACs across species.

9) Figure 7. The authors never indicate the number of independent experiments performed and they should. I guess all experiments in Figure 7 are done with capacitated sperm which should be indicated. Progesterone does not trigger AR in non-capacitated sperm.

We have explicitly stated that AR assays were conducted on capacitated sperms in both Results and Discussion (on Page 12) and in Figure 7 legend (Page 26). The number of independent experiments is now mentioned in Figure 7 legend.

10) In Figure 7C. Why does bicarbonate not stimulate but actually decreases cAMP when it stimulates sAC without peptide, as shown in B? Why does WT peptide stimulate above PBS, it should decrease cAMP?11) In line 277 there is a space between comas that should be deleted.

We have corrected this typo.

12) In Figure 7 why does A23187 induce such a low % of full AR.

It is possible that we see a lower % of complete AR because the concentration of A23187 is 10 µM in our studies for 15 min. Others have used 10 µM for 30 min^2^ or 20 µM for 10 min^3^. We had purposefully chosen this dose and time so that we would be able to test our hypothesis if GIV can exert an inhibitory effect (‘brake’) on stimuli-induced AR. Stimuli that is either too high or too prolonged were therefore avoided.

13) The authors are ignoring work that has indicated mechanisms that control premature acrosome reaction such as:Harper, C. V., Barratt, C.L.R.R., Publicover, S.J., 2004. Stimulation of human spermatozoa with progesterone gradients to simulate approach to the oocyte. Induction of [Ca^2+^]i oscillations and cyclical transitions in flagellar beating. J. Biol. Chem. 279, 46315-46325. https://doi.org/10.1074/jbc.M401194200Balestrini PA, Sanchez-Cardenas C, Luque GM, Baro Graf C, Sierra JM,Hernández-Cruz A, Visconti PE, Krapf D, Darszon A, Buffone MG. Membrane hyperpolarization abolishes calcium oscillations that prevent induced acrosomal exocytosis in human sperm. FASEB J. 2021 Jun;35(6):e21478. doi:10.1096/fj.202002333RR. PMID: 33991146.Sánchez-Cárdenas C, Romarowski A, Orta G, De la Vega-Beltrán JL, Martín-Hidalgo D, Hernández-Cruz A, Visconti PE, Darszon A. Starvation induces an increase in intracellular calcium and potentiates the progesterone-induced mouse sperm acrosome reaction. FASEB J. 2021 Apr;35(4):e21528. doi:10.1096/fj.202100122R

All 3 references have now been added on Page 14, and the reader is informed of these mechanistic insights into sperm-extrinsic factors that regulate premature AR.

14) It is unfair not to indicate that the group of Mayorga in Argentina has described since some years the fundamental role of small G proteins like Rab in the acrosome reaction.

In Figure 1—figure supplement 1 we had cited 4 manuscripts to highlight the role of small GTPases in AR. Mayorga’s work happens to be 2 of the 4 citations.

– Branham MT, Bustos MA, De Blas GA, Rehmann H, Zarelli VE, Treviño CL, Darszon A, Mayorga LS, Tomes CN. Epac activates the small G proteins Rap1 and Rab3A to achieve exocytosis. J Biol Chem. 2009 Sep 11;284(37):24825-39. doi: 10.1074/jbc.M109.015362. Epub 2009 Jun 22. PMID: 19546222; PMCID: PMC2757186.

– Branham MT, Mayorga LS, Tomes CN. Calcium-induced acrosomal exocytosis requires cAMP acting through a protein kinase A-independent, Epac-mediated pathway. J Biol Chem. 2006 Mar 31;281(13):8656-66. doi: 10.1074/jbc.M508854200. Epub 2006 Jan 10. PMID: 16407249.

– Ruete MC, Lucchesi O, Bustos MA, Tomes CN. Epac, Rap and Rab3 act in concert to mobilize calcium from sperm's acrosome during exocytosis. Cell Commun Signal. 2014 Aug 27;12:43. doi: 10.1186/s12964-014-0043-0. PMID: 25159528; PMCID: PMC4156617.

– Lucchesi O, Ruete MC, Bustos MA, Quevedo MF, Tomes CN. The signaling module cAMP/Epac/Rap1/PLCε/IP3 mobilizes acrosomal calcium during sperm exocytosis. Biochim Biophys Acta. 2016 Apr;1863(4):544-61. doi: 10.1016/j.bbamcr.2015.12.007. Epub 2015 Dec 17. PMID: 26704387.

During this revised submission, we have now added 3 more references (on Page 12) to highlight the role of small G proteins in AR.

– Rab27 and Rab3 sequentially regulate human sperm dense-core granule exocytosis.

Bustos MA, Lucchesi O, Ruete MC, Mayorga LS, Tomes CN.Proc Natl Acad Sci U S A. 2012 Jul 24;109(30):E2057-66. doi: 10.1073/pnas.1121173109. Epub 2012 Jul 2.PMID: 22753498

– Small GTPases in acrosomal exocytosis.

Bustos MA, Lucchesi O, Ruete MC, Mayorga LS, Tomes CN.Methods Mol Biol. 2015;1298:141-60. doi: 10.1007/978-1-4939-2569-8_12.PMID: 25800839

– ADP ribosylation factor 6 (ARF6) promotes acrosomal exocytosis by modulating lipid turnover and Rab3A activation.

Pelletán LE, Suhaiman L, Vaquer CC, Bustos MA, De Blas GA, Vitale N, Mayorga LS, Belmonte SA.J Biol Chem. 2015 Apr 10;290(15):9823-41. doi: 10.1074/jbc.M114.629006. Epub 2015 Feb 20.PMID: 25713146

15) Were mouse epidydimal sperm collected by swim out or swim up, please indicate?

Mouse epidydimal sperm was collected by swim out. In this revised submission, this point has been explicitly states, and details surrounding the steps during sperm collection have clarified by expanding the methods sub-section entitled “Source of live mouse and human sperms”.

16) Please indicate the progesterone and A23187 concentrations used.

We have now corrected this error of omission by adding the information in both text and legend.

17) Line 414 should be were not where.

We have now corrected this error.

18) In human sperm capacitation for 3 hours is barely enough.

We collected ejaculated sperm from human subject volunteers. For all human sperm studies, we used 30 min and 4 h (not 3 h). These time points were chosen because the incubation time for capacitation of human sperm in vitro is believed to range from ~3-24 h^4^.

Reviewer #3 (Recommendations for the authors):It is not clear how to interpret the absence of GIV transcripts from "Sertoli Cell Only" testes. These testes lack all germ cells. Absence of GIV confirms that GIV is expressed in male germ cells but says little about the role of GIV in fertility or sperm function. It may make sense to recalculate the significance in Figure 2E, excluding the Sertoli cell only samples to focus on samples from men who make defective sperm.

We agree with the reviewer that Sertoli cells only (SCO), a condition that lacks germ cells, is not the best sample to study the impact of low GIV levels in sperm on male fertility.

Actions taken: We have now replaced the original panels in Figure 2E with a new analysis that represents pre-pubertal Klinefelter’s and adult Klinefelter’s syndrome (KS), after excluding Sertoli cell only (SCO) subjects. But to avoid picking and choosing subjects from the cohort, we included the SCO subjects as a positive control, as was intended in the study. These new analyses show that levels of GIV (CCDC88A) transcripts drop in Klinefelter’s and Klinefelter-like syndromes *after* puberty, but not in pre-pubertal subjects. This finding is in keeping with the clinical observation that germ cells are depleted in patients with this condition at the onset of puberty [J Clin Endocrinol Metab. 2004 May;89(5):2263-70. doi: 10.1210/jc.2003-031725. PMID: 15126551. Link: https://pubmed.ncbi.nlm.nih.gov/15126551/].

As predicted by this reviewer, we confirmed that the positive controls used in this study to recapitulate gamete depletion (i.e., SCO patients) have significantly low GIV in the absence of gametes. (see Figure 2E; 4 subjects in each group). The figure legend has been edited accordingly.

The word "sperm" appears in the Impact Statement and should be replaced with "spermatozoa".

Done, as recommended.